# Chaperone BiP controls ER stress sensor Ire1 through interactions with its oligomers

Sam Dawes[1,2,*], Nicholas Hurst[1,*], Gabriel Grey[1], Lukasz Wieteska[1], Nathan V Wright[1], Iain W Manfield[1], Mohammed H Hussain[1], Arnout P Kalverda[1], Jozef R Lewandowski[3], Beining Chen[2], Anastasia Zhuravleva[1]

**The complex multistep activation cascade of Ire1 involves changes in the Ire1 conformation and oligomeric state. Ire1 activation enhances ER folding capacity, in part by overexpressing the ER Hsp70 molecular chaperone BiP; in turn, BiP provides tight negative control of Ire1 activation. This study demonstrates that BiP regulates Ire1 activation through a direct interaction with Ire1 oligomers. Particularly, we demonstrated that the binding of Ire1 luminal domain (LD) to unfolded protein substrates not only trigger conformational changes in Ire1-LD that favour the formation of Ire1-LD oligomers but also exposes BiP binding motifs, enabling the molecular chaperone BiP to directly bind to Ire1-LD in an ATP-dependent manner. These transient interactions between BiP and two short motifs in the disordered region of Ire1-LD are reminiscent of interactions between clathrin and another Hsp70, cytoplasmic Hsc70. BiP binding to substrate-bound Ire1-LD oligomers enables unfolded protein substrates and BiP to synergistically and dynamically control Ire1-LD oligomerisation, helping to return Ire1 to its deactivated state when an ER stress response is no longer required.**

## Introduction

Folding and maturation of the most secreted and membrane proteins occur in the ER with the assistance of the ER protein quality control (PQC) networks. The protein folding capacity in the ER must be efficiently adjusted for the cell to react to physiological and pathological changes in the ER protein load and to ensure correct protein folding. The balance between the protein load and the ER folding capacity relies on a regulatory mechanism called the unfolded protein response (UPR) (1, 2, 3). If the folding capacity is exhausted and unfolded and misfolded proteins start to accumulate in the ER, three UPR sensors, protein kinase R-like ER kinase (PERK), activating transcription factor 6 (ATF6), and inositol-requiring enzyme 1 (Ire1), are activated. The activation of these three branches results in the overexpression of ER chaperones and other components of the PQC network to restore the ER proteostasis (4, 5).

Ire1 represents the most conserved of ER stress sensors. Within the Ire1 family, there are two Ire1 paralogues: Ire1α (referred to as Ire1 in this study), which is ubiquitously expressed, and Ire1β, which is specifically expressed in epithelial cells (6). Ire1 becomes active when its luminal domain (LD) senses ER stress (7). Moreover, growing evidence suggests that the activation process is directed by the luminal domain (8). Upon accumulation of unfolded proteins in the ER, Ire1 luminal domain oligomerises, resulting in consequent oligomerisation of the cytoplasmic domain (CD), triggering, in turn, Ire1-CD auto-transphosphorylation and activation (9). The active Ire1 non-conventionally slices mRNA of a transcription factor X-box binding protein 1 (10), resulting in its stable form that up-regulates ER chaperones and PQC enzymes, which act to resolve ER stress (4, 5).

Despite significant progress in the structural characterization of the Ire1 activation cascade, our knowledge about the initial step of Ire1 activation, oligomerisation of Ire1-LD, remains controversial and incomplete. Whereas reversible dimerization and consequent oligomerisation of Ire1-LD has been widely accepted to be an essential step for activation of the protein (11, 12, 13), currently, two alternative models have been suggested to explain how the accumulation of unfolded proteins in the ER triggers changes in the Ire1-LD oligomeric state. The first model relies on unfolded protein substrates *directly* interacting with constitutive inactive Ire1-LD homodimers, resulting in conformational changes in Ire1-LD that propagate to the Ire1-LD oligomerisation interface and, as a result, favours Ire1-LD oligomerisation (13, 14). The second model proposes that unfolded proteins only *indirectly* control Ire1 activation by competing with Ire1-LD for binding to the ER Hsp70 molecular

[1]School of Molecular and Cellular Biology, Faculty of Biological Sciences & Astbury Centre for Structural Molecular Biology, University of Leeds, Leeds, UK   [2]Chemistry Department, University of Sheffield, Sheffield, UK   [3]Department of Chemistry, University of Warwick, Coventry, UK

Correspondence: a.zhuravleva@leeds.ac.uk
Nicholas Hurst's present address is Proimmune Ltd, Magdalen Centre, Oxford, UK
Lukasz Wieteska's present address is Developmental Signalling Laboratory, The Francis Crick Institute, London, UK
Nathan V Wright's present address is Queen Mary University of London, London, UK
*Sam Dawes and Nicholas Hurst contributed equally to this work

chaperone BiP (Binding immunoglobulin Protein) ([15], [16]). In this model, BiP plays a key role in controlling Ire1-LD oligomeric states. BiP binds to Ire1-LD in the absence of ER stress (accumulation of unfolded proteins), preventing Ire1-LD oligomerisation, but dissociates from Ire1-LD upon interactions with the excess of unfolded proteins.

Furthermore, how exactly BiP interacts with Ire1-LD is under much debate ([16], [17], [18]). Earlier observations proposed interactions between Ire1-LD and BiP in the absence of ER stress ([19]); moreover, the C-terminus unstructured region of yeast ([20], [21]) and human ([22]) Ire1-LD has been shown to be essential for interactions with BiP. The recent elegant studies by Amin-Wetzel et al ([15], [18]) have revealed some molecular details of interactions between human Ire1-LD and BiP, demonstrating that BiP association with Ire1-LD is transient and requires the assistance of BiP co-chaperone ERdj4. According to these studies, Ire1-LD interacts with the BiP substrate-binding domain and these interactions are under the control of ATP hydrolysis in the BiP nucleotide-binding domain. These chaperone–substrate–like interactions disrupt Ire1-LD activation by disfavouring the Ire1-LD dimer and stabilizing its monomer. Alternately, it has also been suggested that BiP interacts with Ire1 through its nucleotide-binding domain (BiP NBD) in a non-canonical, ATP-independent manner ([23]), whereas BiP binding to its unfolded protein substrate perturbs these interactions ([24]) through a yet unknown allosteric mechanism ([16]).

Growing evidence suggests that both direct binding of unfolded protein substrates to Ire1-LD and interactions between Ire1-LD and BiP are likely to play important and distinctive roles in Ire1 activation and its regulation ([5]). However, it remains obscure whether BiP only attenuates early steps of Ire1-LD signalling through controlling the population of Ire1-LD dimers, which are essential for unfolded substrate-binding and consequent oligomerisation ([17], [25], [26]) or if it plays any active role in the disassembly of active Ire1-LD oligomers as well. In this study, we addressed this question by elucidating how Ire1-LD interactions with BiP and unfolded protein substrates cross-talk with each other to regulate the Ire1-LD oligomerisation process. We performed a detailed biophysical characterization of how the molecular chaperone BiP and a model Ire1 substrate interdependently reshape the Ire1-LD functional landscape and, by doing this, mutually control Ire1-LD activation, enabling precise fine-tuning of the Ire1 activation process.

## Results

### Substrate binding to Ire1-LD results in multistep, dynamic oligomerisation

To characterize the initial steps of the Ire1 activation process, we first performed the biophysical characterization of the luminal domain (LD) of human Ire1$\alpha$ (called Ire1-LD in this study). Particularly, we elucidated how the presence and absence of its model substrate affect the Ire1-LD oligomeric state. We used microscale thermophoresis (MST), size exclusion chromatography (SEC) (Figs 1A, S1, and S2A and B), and native mass spectrometry (MS, Fig S3) to characterize the oligomeric state of apo Ire1-LD. In agreement with previous observations ([23]), MST, SEC, and MS consistently demonstrated the formation of dimer and a small fraction of tetramers in the absence of protein substrates. The apparent dimerization constant of apo wild-type Ire1-LD detected by MST and SEC was in a sub-$\mu$M range (0.5 ± 0.17 $\mu$M, Fig 1A).

To monitor how binding to an unfolded protein substrate affects the Ire1-LD oligomeric state (and thus its activation), we used two previously characterized model peptides ΔEspP ([14]) and MPZ1 ([13]) (apparent $K_{1/2}$ of binding of 6.4 $\mu$M ([23]) and 24 $\mu$M ([13]), respectively). These model peptides have been previously used to probe substrate-induced oligomerisation of Ire1-LD that comprises several steps: substrate binding to Ire1-LD dimer, its conformational changes, and consequent formation of Ire1-LD oligomers ([13]). In line with previous results, our dynamic light scattering measurements demonstrated that the addition of ΔEspP resulted in the formation of soluble Ire1-LD oligomers (Fig 1B).

Interestingly, the size of Ire1-LD oligomers depends on peptide concentration and/or substrate affinity from the formation of relatively small oligomers at low peptide concentrations (Fig 1B) to slow (>30 min) formation of insoluble oligomers at higher peptide concentrations (Figs 1C and S4A–D). The apparent constant of the ΔEspP-induced formation of soluble Ire1-LD oligomers measured by fluorescence polarisation (at low nM Ire1-LD concentrations) was in good agreement with the formation of insoluble oligomers observed at higher ($\mu$M) Ire1-LD concentrations (see Fig S5A and B). In turn, the MPZ1 peptide that binds to Ire1-LD with a significantly lower affinity than ΔEspP (6.4 ([23]) versus 24 $\mu$M ([13])), required significantly higher concentrations to promote the formation of insoluble Ire1-LD oligomers (Fig S6), revealing a clear correlation between the peptide binding and formation of soluble and insoluble oligomers.

To further examine whether the substrate-induced formation of the insoluble Ire1-LD oligomers has the same mechanism as the formation of soluble, lower order oligomers, we used previously characterized variants of Ire1-LD that contain either the D123P ([12]) mutation at the dimerization interface or the [359]WLLI[362] to GSSG substitution at the oligomerisation interface ([13]). It has been previously shown that both mutations drastically suppress Ire1 activation in vivo and the formation of soluble oligomers in vitro ([12], [13]). In our hands, both variants not only suppressed the formation of soluble Ire1-LD oligomers but also showed a drastically reduced ability to form large, insoluble Ire1-LD oligomers (Fig S7A–C).

Solid-state NMR and transmission electron microscopy further validated that these large insoluble oligomers have structural features of functional (soluble) Ire1-LD oligomers. Particularly, the solid-state amide NMR spectrum of insoluble [15]N-labelled Ire1-LD.ΔEspP oligomers had characteristic features of folded proteins (Fig S8A), suggesting that in the insoluble oligomeric form, Ire1-LD adopted its folded conformation. Furthermore, negative-stained transmission electron microscopy of the insoluble Ire1-LD.ΔEspP oligomers revealed the formation of fibril-like structures with a diameter consistent with the size of functional Ire1-LD dimers (Figs 1D and S8B and C). Interestingly, the in vitro fibrils observed in our study are closely reminiscent of Ire1-LD filaments that have been recently observed in vivo under stress conditions ([28]). Altogether, these results suggest that the insoluble Ire1-LD oligomers trapped (at least) in vitro at high Ire1-LD and/or peptide concentrations are

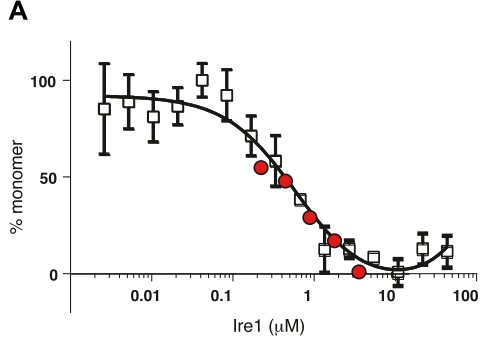

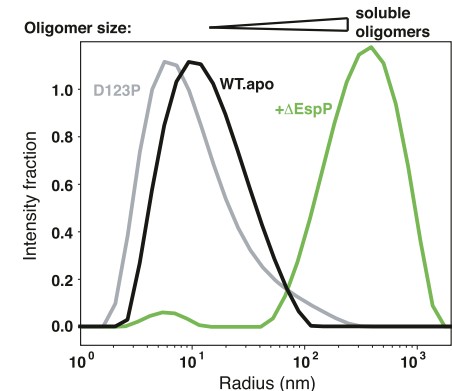

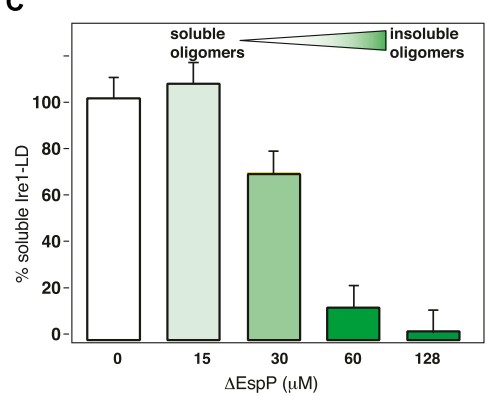

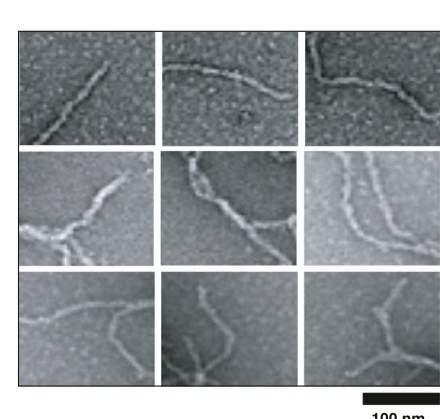

100 nm

**Figure 1. Ire1-LD binding to unfolded protein substrate results in its oligomerization.**

**(A)** The oligomerization state of apo Ire1-LD monitored by MST and SEC. The fraction of monomeric Ire1-LD as a function of the Ire1-LD concentration was plotted using normalized MST (empty squares, Fig S1) and SEC (red circles, Fig S2) data. The black line represents the best fit of MST data (with the apparent dimerization constant $K_{1/2}$ of 0.5 ± 0.17 $\mu$M). Error bars indicate ±SE for three replicate experiments. **(B)** The oligomerization of 5 $\mu$M Ire1-LD in the absence (in black) and in the presence (in green) of the 10 $\mu$M ΔEspP monitored by DLS (regularisation plots are shown). The monomeric peak from the dimerization-deficient D123P variant (12) of Ire1-LD is shown in grey. **(C)** Formation of large insoluble Ire1-LD oligomers monitored by the solubility assay; the experiments were performed for 20 $\mu$M Ire1-LD in the presence of the different (0–128 $\mu$M) concentrations of ΔEspP. **(D)** TEM image of elongated Ire1-LD oligomers obtained in the presence of 128 $\mu$M ΔEspP (same as for Fig 1C, dark green).

an extended form of soluble Ire1-LD oligomers and, thus, provide a valuable probe to monitor the Ire1-LD activation process.

### The molecular chaperone BiP controls Ire1-LD oligomerisation in a chaperone-like ATP-dependent manner

Next, we examined whether BiP directly affects the formation of Ire1-LD oligomers induced by binding to the ΔEspP peptide. Because BiP does not interact with the model peptide ΔEspP (23) and, thus, does not compete with Ire1-LD for ΔEspP binding, any changes in Ire1-LD oligomerisation in the presence of BiP observed in our experiments were because of the direct cross-talk between BiP and Ire1-LD. We found that BiP drastically reduced the size of soluble oligomers (Figs 2A and S9A) and resulted in the solubilisation of insoluble Ire1-LD oligomers (Fig 2B, left). Remarkably, the BiP-driven de-oligomerisation of Ire1-LD was only observed in the presence of ATP and required active BiP. No substantial solubilisation of Ire1-LD fibrils was observed in the absence of ATP or with inactive BiP variants (Fig 2B, middle) that contain either the T229G or V461F substitutions, which compromise ATPase activity (T229G (29)) and substrate binding (V461F (30)). Neither excess of BiP nor its equimolar concentration is required for Ire1-LD de-oligomerisation as the effect of BiP was observed even at the BiP:Ire1-LD ratio of 1:100 (pink in Figs 2B and S9B), suggesting that the formation of a stable one-to-one complex between BiP and Ire1-LD is not required for the BiP-driven Ire1-LD de-oligomerisation process. These data, therefore,

suggest that Ire1-LD de-oligomerisation relies on transient ATP-dependent interactions with BiP.

No stable complex formation between dimeric (inactive) Ire1-LD and BiP was observed by methyl NMR (Fig S10). However, in the absence of ATP, BiP co-precipitated with Ire1-LD.ΔEspP oligomers (Fig 2C) but did not result in Ire1-LD de-oligomerisation. In turn, the addition of ATP triggered solubilisation of both Ire1-LD and BiP (Fig 2D). Notably, BiP solubilised only folded Ire1-LD.ΔEspP oligomers but not heat-denatured (misfolded) Ire1-LD aggregates (Fig 2B, right), indicative of BiP acting specifically on the folded (functional) form of Ire1-LD. Altogether, these results suggest that BiP acts as a chaperone that transiently binds to folded Ire1-LD oligomers and uses the energy of ATP hydrolysis to de-oligomerise them.

### BiP binds to two distinct sites at the Ire1-LD oligomerisation interface

The flexible C-terminal region of Ire1-LD (residues H301-S449) has been previously suggested to bind BiP(18) and be important for Ire1-LD oligomerisation and clustering (13) (8, 31 Preprint); however, the exact BiP binding site(s) has not been identified. We used the state-of-the-art BiPPred algorithm (32) to predict potential BiP binding motifs in this flexible C-terminal region. To then experimentally validate whether the BiPPred-predicted motifs bind to BiP, we produced 7- or 8-residue peptides (Table S1) and recorded methyl NMR spectra of ATP-bound methyl-labelled BiP in the presence and the absence of the unlabelled peptides.

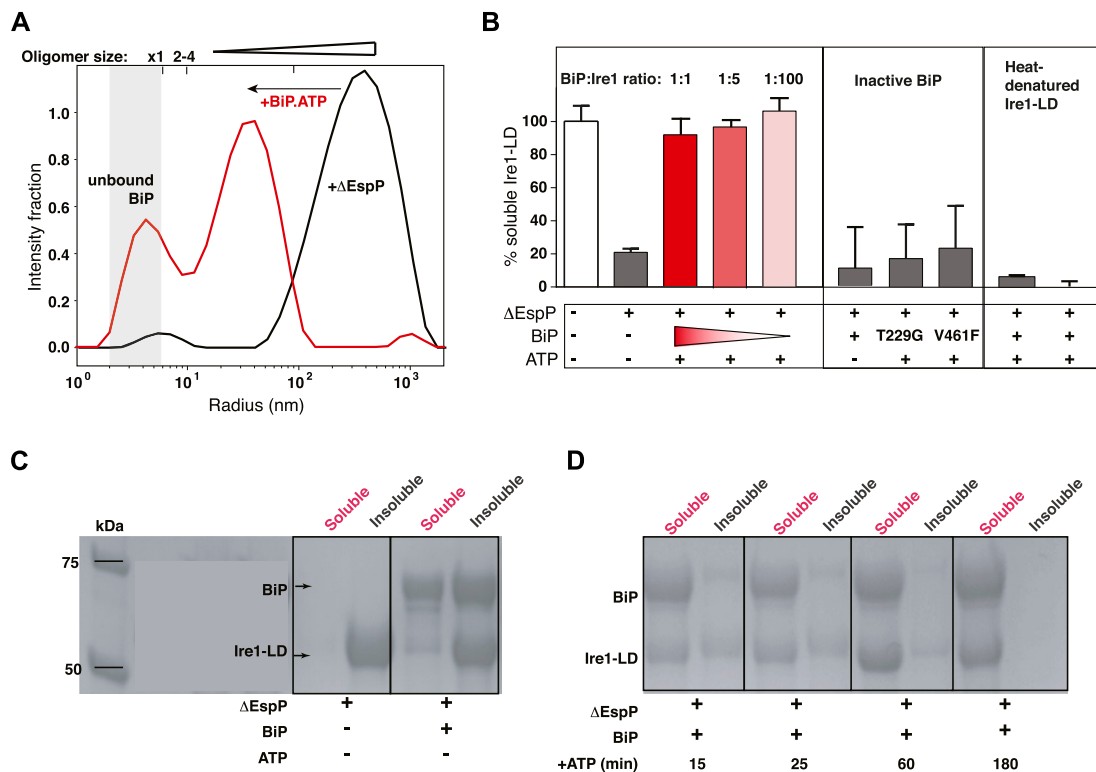

**Figure 2. BiP interacts with Ire1-LD oligomers in a chaperone-like manner.**
**(A)** BiP reduces the size of soluble oligomers. The DLS measurements were performed using 5 µM Ire1-LD in the presence of 10 µM ΔEspP and in the absence (black, same as for Fig 1B) and presence (red) of 5 µM BiP.ATP. **(B)** BiP solubilises large insoluble Ire1-LD oligomers as monitored by the solubility assay. The solubility assay was performed on 20 µM Ire1-LD in the absence and presence of 170 µM ΔEspP. Error bars represent standard deviations. At these concentrations, the peptide induces the formation of insoluble Ire1-LD oligomers, whereas adding BiP and ATP results in solubilisation of these oligomers (left, red). Sub-stoichiometric BiP concentrations are sufficient to solubilised insoluble Ire1-LD oligomers (left, pink); no de-oligomerisation occurs in the absence of ATP (left, grey) or in the presence of chaperone-inactive variants of BiP, which compromise ATPase activity (T229G (29)) and substrate binding (V461F (30)) (middle), suggesting that BiP disturbs Ire1-LD oligomerisation in a chaperone-like manner. Moreover, BiP does not affect misfolded Ire1-LD aggregates obtained by heat denaturation (right). **(C)** In the absence of ATP, BiP co-precipitates with Ire1-LD insoluble oligomers. The SDS–PAGE gel shows the soluble and insoluble fractions of Ire1-LD ΔEspP samples in the presence and in the absence of 20 µM BiP in the absence and presence of ATP. **(D)** The addition of ATP results in BiP dissociation from Ire1-LD insoluble oligomers and their solubilisation. The SDS–PAGE gel shows the soluble and insoluble fractions of Ire1-LD ΔEspP samples in the presence 20 µM BiP at different times (15–180 min) after the addition of 40 mM ATP. 20 µM Ire1-LD was incubated with 170 µM ΔEspP for 3 h before the addition of ATP.

The methyl NMR demonstrated that only two Ire1-derived peptides, [310]GSTLPLLE[317] and [356]RNYWLLI[362], behave as 'classical' BiP substrates (Fig 3A). Binding to these peptides resulted in substrate-induced conformational changes in ATP-bound BiP, similar to ones previously observed for the BiP model substrate HTFPAVL (33). Particularly, in the absence of the peptide substrates, ATP-bound BiP co-exists as an ensemble of two functional states, domain-docked and -undocked, as monitored by two sets of NMR signals; the addition of the peptides resulted in the redistribution of its conformational ensemble, favouring its domain-undocked conformation (Figs 3A and S11A–C). In contrast to the [310]GSTLPLLE[317] and [356]RNYWLLI[362] peptides, the other four BiPPred-predicted peptides result in no significant perturbations in the BiP NMR spectrum, indicative of no binding (Fig S11A–C). These findings suggest that the [310]GSTLPLLE[317] and [356]RNYWLLI[362] motifs from the C-terminal region of Ire1-LD behave as classical BiP substrates, that is, bind to the BiP substrate-binding site in the ATP-dependent manner. As expected for canonical Hsp70 substrates (34), the addition of either [310]GSTLPLLE[317] and [356]RNYWLLI[362] peptides results in enhanced ATPase activity of WT BiP, comparable with the effect observed in the presence of the model substrate HTFPAVL (32, 33) (Fig 3B). Interestingly, whereas perturbations in the [310]GSTLPLLE[317] and [356]RNYWLLI[362] region in the full-length Ire1-LD considerably affect Ire1-LD oligomerisation and clustering of Ire1-LD (13) (8, 31 Preprint), perturbations in either motif were insufficient to eliminate Ire1-LD oligomerization and BiP-induced de-oligomerisation completely (Fig S12), indicative of the multivalent nature of these interactions between Ire1-LD oligomers and BiP.

**BiP binding sites are located in a highly dynamic oligomerization interface of Ire1-LD**

Despite the functional importance of the C-terminal part of Ire1-LD for oligomerisation and BiP binding (as demonstrated in this and previous studies (8, 13, 18, 31 Preprint)), to date, only very limited information is available about the last 150 residues of Ire1-LD (residues 301–449). Given that most of this region lacks electron density (PDB IDs 2HZ6 (12) and 6SHC (18)) and is reported to be highly dynamic (18), a deeper understanding of how this region orchestrates BiP binding and oligomerisation is needed. To

▶▶▶▶ Life Science Alliance

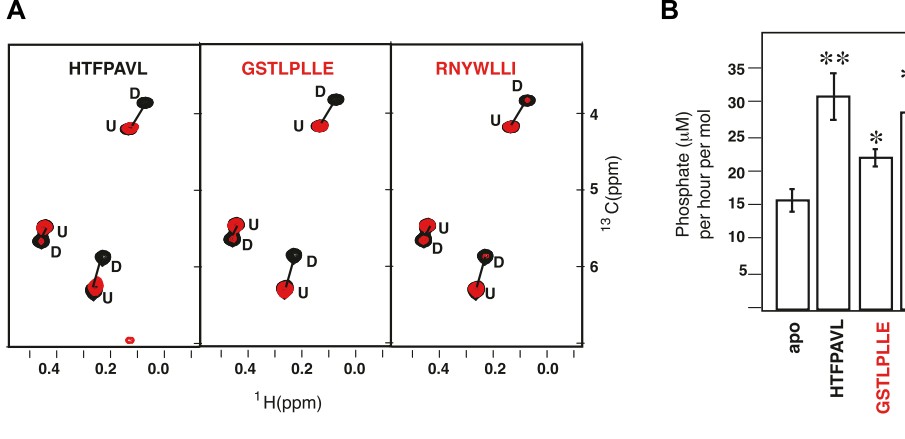

**Figure 3.   BiP binds to two distinct disordered motifs in Ire1-LD.**
**(A)** Two Ire1-derived peptides result in substrate-like perturbations in the BiP conformational landscape: The representative region of the methyl-TROSY spectra of ATP-bound full-length U{$^2$H,$^{12}$C}, Ile-C$\delta^1$-$^{13}$CH$_3$ BiP* T229G in the absence (black) and the presence of 7 aa peptides (red): the model BiP substrate HTFPAVL (left) and two Ire1-derived BiP-binding peptides GSTLPLLE (middle) and RNYWLLI (right). No changes in spectra were observed in the presence of the other Ire1-derived peptides (Fig S11). **(B)** The GSTLPLLE and RNYWLLI peptides result in the substrate-like stimulation of the ATPase activity in BiP. The ATPase activity of 1 μM BiP* was measured in the absence of any substrate and the presence of 1 mM of either the model BiP substrate HTFPAVL or either GSTLPLLE and RNYWLLI peptide. Error bars represent standard deviations. The asterisk (*) represents the P-value of the statistical test. *P < 0.05, **P < 0.001.

investigate this functionally important region, we used solution NMR and the sequence conservation analysis combined with the AlphaFold structural prediction.

Based on the sequence conservation analysis (Fig S13A and B), the C-terminal part of Ire1-LD (residues 301–449) can be subdivided into two parts: the highly conserved part (residues 301–390), which is adjacent to the core Ire1-LD (residues 24–300) and the significantly less conserved juxtamembrane region (residue 391–449). The truncation of the juxtamembrane part of this region only slightly affected the protein's stability (Fig S14A and B), its interactions with BiP (18), and stress-induced oligomerisation of Ire1-LD (13). Most (57 from 59 expected) of the peaks from the juxtamembrane region were present in the amide NMR spectrum; these peaks were highly intense with the low proton dispersion (Fig S15A and B) and temperature gradients typical for unstructured regions (Fig S15C), which agrees with the predictions that this region is disordered (Fig S16). In contrast, most of the highly conserved region adjacent to the core Ire1-LD (residues 301–390) is not predicted to be intrinsically disordered (Fig S16). Moreover, both core Ire1-LD and residues 301–390 adjacent to the core were predominantly "invisible" or their peak intensities are significantly lower than ones for the juxtamembrane region (Fig S15B). These results suggest that residues 24–390 are affected by conformational dynamics on the μs-ms timescale resulting in drastic peak broadening and low intensities for most of the residues in Ire1-LD. In contrast, the juxtamembrane region (residues 391–449) is predominantly unaffected by this conformational flexibility.

The analysis of AlphaFold-based structural models for different members of Ire1 family provides a plausible explanation for the μs-ms conformational flexibility observed by NMR. In most of the modelled AlphaFold structures, residues 301–390 exhibit very low AlphaFold confidence scores (pLDDT < 50) and are mainly disordered, with only two short β-strands (corresponding to $^{306}$VVP$^{308}$ and $^{363}$GHH$^{365}$ in human Ire1α) adjacent to the β-hairpin of the core Ire1-LD (residues 281–300 in human Ire1) (Fig 4A, middle). Although these same two β-strands are present in the X-ray structures of human Ire1-LD (12, 18) (Fig S16) and were consistently predicted across most Ire1 homologues, the confidence scores for these predictions were relatively low (70> pLDDT >50), suggesting

conformational flexibility in this region. Interestingly, in *Drosophila* Ire1s, the formation of an additional β-strand ($^{333}$VIT$^{335}$ in human Ire1α) in this region (Fig 4A, left), results in significantly higher AlphaFold confidence scores (pLDDT > 70), suggesting that the formation of this additional β-strand reduces conformational flexibility this region. Importantly, the three β-strands are highly conserved (Fig 4B) in the Ire1 family, indicative of their structural significance not only for *Drosophila* Ire1s but for the entire family. In several Ire1 homologues, neither these residues nor the adjacent region corresponding to the β-hairpin of the core Ire1-LD (residues 281–300 in human Ire1) adopts secondary structure (Fig 4, right), revealing an alternative (unfolded) conformation for residues 301–390. Taken together, these findings suggest that the Ire1-LD conformational flexibility can be attributed to folding-unfolding transitions at the C-terminal oligomerisation interface.

Both BiP binding sites are adjacent to the central β-strands of the oligomerization interface ($^{306}$VVP$^{308}$ and $^{363}$GHH$^{365}$ in human Ire1, Fig 4A and B), suggesting that interactions with BiP and structural rearrangements within this region are interdependent. Interestingly, subtle sequence variations in the $^{356}$RNYWLLI$^{362}$ motif among different Ire1-LD homologues do not significantly alter BiP binding, as predicted by the BiPPred algorithm (32) (Table S2). In turn, the conservation analysis identified two highly conserved variants of the $^{310}$GSTLPLLE$^{317}$ motif: one predominantly found in Ire1α sub-family (Fig 4C, red) and the other in the Ire1β sub-family (Fig 4C, blue); both motifs are predicted to bind to BiP (Table S2). Overall, our results suggest that the BiP binding sites are evolutionarily conserved, emphasizing the significance of BiP-driven de-oligomerization in the Ire1 activation cascade.

## Discussion

We investigated whether and how BiP controls the critical steps of Ire-LD activation: stress-dependent assembly and disassembly of Ire1-LD oligomers. Whereas the size of oligomers remains a debatable question and is likely to depend on a number of factors such as Ire1 concentrations and the level of stress, their importance

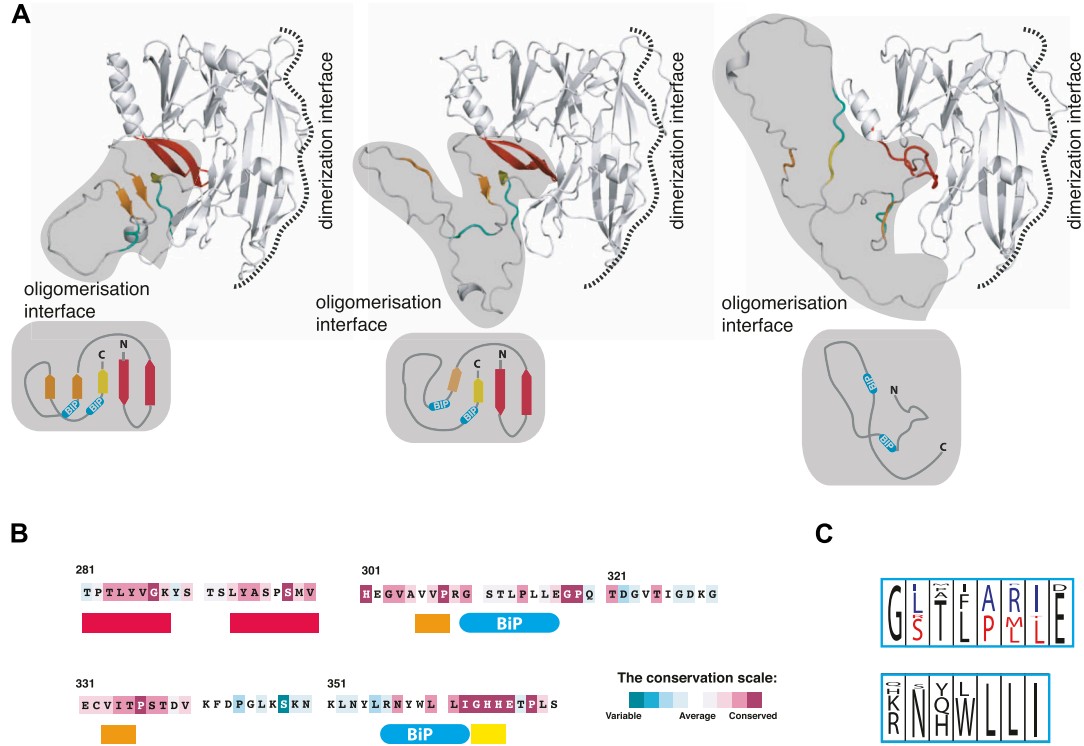

**Figure 4. Conformational rearrangements in the C-terminal oligomerisation subdomain of Ire1-LD.**
**(A)** Three representative AlphaFold models of Ire1 luminal domain are shown, depicting the region corresponding to residues 24–365 in human Ire1, to illustrate three distinct conformations of the C-terminal subdomain of Ire1-LD. The oligomerization interface is shaded in grey for clarity; the β-strands [306]VVP[308] and [333]VIT[335] (as in human Ire1) are highlighted in orange; the [363]GHH[365] β-strand is highlighted in yellow; the adjacent β-hairpin (residues 281–300) is shown in red; two BiP binding sites, [310]GSTLPLL[316] and [356]RNYWLLI[362] in human Ire1, are depicted in cyan and labelled. The dimerization interface, as suggested previously (12, 13), is annotated. The AlphaFold structures used for this comparison are AF-A8JR46 (*Drosophila melanogaster*, left), AF-A0A7P0TAB0 (*Homo sapiens*, middle), and AF-A0A668SMD8 (*Oreochromis aureus*, right). The cartoons at the bottom represent three different conformations of the oligomerization subdomain. **(B)** The oligomerization interface of Ire1-LD is highly conserved, as evidenced by Consurf conservation scores. Briefly, a total of 124 unique sequences, including Ire1α and Ire1β paralogues from metazoans (6) (Fig S13), were selected and analysed using the ConSurf server. The highest conservation scores were observed for β-strands [306]VVP[308], [333]VIT[335], and [363]GHH[365], and the adjacent β-hairpin (highlighted by the same colours as in (A)). **(A, B, C)** Sequence conservation logos for the BiP binding motifs [310]GSTLPLLE[317] and [356]RNYWLLI[362] are shown in cyan as in (A, B). The paralogue-specific amino acid conservation, representing amino acid types found exclusively in the Ire1α and Ire1β paralogues (6), is denoted in red (Ire1α) and blue (Ire1β); amino acid types found in the same position across the entire family without paralogue-specific conservations are represented in black. The conservation level, as calculated in iTOL (37), is indicated by the height of the symbols representing amino acid types with the one-letter code used for representation.

for the Ire1 activation process has been shown both in vitro and in vivo (8, 13). In this study, we found that BiP directly interacts with Ire1-LD oligomers in a chaperone-like manner, providing an additional level of control for the Ire1 activation cascade. Particularly, the BiP binding results in destabilization of Ire1 oligomers and their gradual disintegration. This mechanism is distinct from two previously suggested mechanisms by which BiP controls the equilibrium between inactive Ire1-LD monomeric and dimeric species in the absence of stress (16): non-canonical interactions with NBD BiP (23, 46) and chaperone-client-like interactions with apo Ire1-LD inactive dimers in the presence of the BiP co-chaperone ERdj4 (15, 18). We found that interactions with active Ire1-LD oligomers do not require ATP; however, their de-oligomerisation is only possible in the presence of ATP and when BiP has both ATPase and substrate-binding activities. Whereas these interactions do not require the assistance of BiP co-chaperones in vitro, co-chaperones, such as ERdj4 (15, 18) and Sec63 (47), are likely to be essential under in vivo conditions, at drastically lower concentrations of Ire1 (few nM in cells); co-chaperones are also likely to provide an additional level of control of the Ire1-LD oligomerisation process through their fine-tuning of binding and release of BiP to their substrates (including Ire1-LD oligomers) but also through delivering Ire1-LD to BiP as previously suggested (15, 18, 47).

Furthermore, our results answer the question of how BiP interacts with fully folded Ire1-LD oligomers in a chaperone-like manner (i.e., solvent-exposed hydrophobic region of Ire1-LD) and how these interactions interfere with dynamic Ire1-LD oligomerisation. Notably, Ire1-LD binding to unfolded protein substrates is a critical initial step for interactions between Ire1 and BiP. Indeed, conformational changes in Ire1-LD, which activate Ire1 favouring its oligomerisation (13), are also essential for interactions with BiP, thus enabling a negative feedback loop. We identified two evolutionary conserved motifs in Ire1 ([310]GSTLPLLE[317] and [356]RNYWLLI[362] in human Ire1) responsible for Ire1 interactions with BiP but also for oligomerisation of Ire1-LD. Intriguingly, these BiP-binding motifs are located in the highly dynamic, oligomerisation interface of Ire1-LD. We speculate that structural rearrangements at this interface play a central role in Ire1-LD activation. ER stress (binding of unfolded protein substrates) triggers structural changes at the oligomerisation interface, altering the availability of

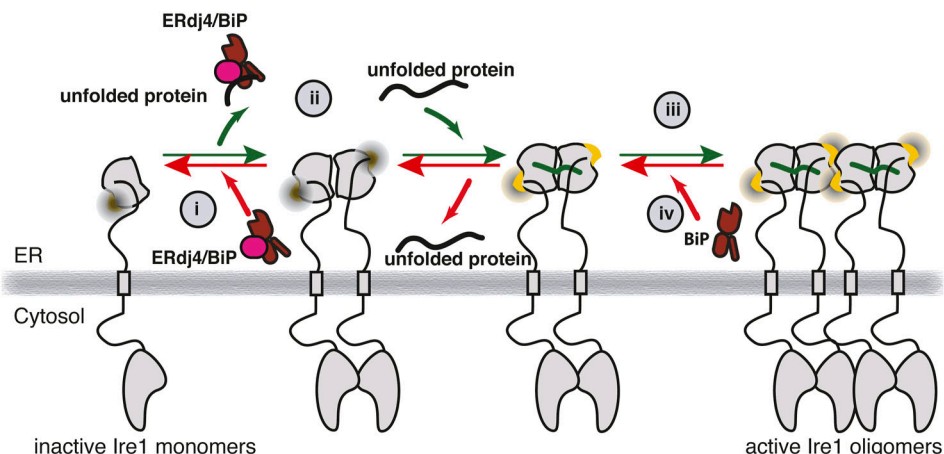

**Figure 5. Multivalent regulation of Ire1 activity by molecular chaperone BiP.**
(i) In the absence of stress Ire1-LD co-exists in an equilibrium between inactive monomers and dimers. With the assistance of its co-chaperone ERdj4, BiP binds to Ire1-LD dimers, favouring monomers (15, 18). (ii) Accumulation of unfolded proteins results in BiP dissociation from the Ire1-LD (15, 18), enabling Ire1 dimerization. In addition, unfolded proteins bind to the Ire1-LD dimers, resulting in conformational changes in the Ire1-LD oligomerization interface (yellow) (13). (iii) Conformational changes at the oligomerization interface result in Ire1-LD oligomerization and consequent activation of Ire1 cytoplasmic domain, which leads to overexpression of ER protein quality control enzymes, including the molecular chaperone BiP (5, 17). (iv) The same stress-induced conformational changes in Ire1-LD enable BiP interactions near the oligomerization interface and, thus, provide additional control for the Ire1-LD activation process. Upon reduction in stress level, when the chaperone becomes available, BiP transiently "bites" the oligomerization interface of Ire1-LD, destabilizing Ire1 oligomers and, thus, facilitating Ire1 deactivation.

BiP-binding and oligomerisation-prone hotspots promoting both oligomerisation and BiP binding. Interestingly, both BiP-binding sites are located in the highly flexible C-terminal part of the Ire1-LD that has been previously suggested to be key for interactions between BiP and Ire1-LD dimer in the absence of unfolded protein substrate (15, 18), opening a possibility that the same regions may be responsible for BiP interactions both in the presence and in the absence of unfolded protein substrate. Interestingly, in the absence of substrate (and Ire1-LD oligomers) interactions between Ire1-LD and BiP require the assistance of the BiP co-chaperone ERdj4, suggesting that by a yet unknown mechanism, ERdj4 could affect the availability of these sites for BiP binding that are not otherwise available in the absence of the unfolded protein substrate.

BiP-driven de-oligomerisation of Ire1-LD relies on the stress-controllable accessibility of the binding hotspots located at the oligomerisation interface, and, thus, is reminiscent of clathrin-coat disassembly mediated by cytoplasmic Hsp70 called Hsc70. Similar to BiP-Ire1 interactions, Hsc70 interacts with the QLMLT motif in the dynamic region of clathrin heavy chain (48) that becomes transiently exposed upon oligomerisation-induced conformational changes, enabling recognition by Hsc70 (49). Similarly, interactions between the cytoplasmic Hsp70 and glucocorticoid receptor rely on a local unfolding of a specific region near the ligand binding pocket of the receptor (50). A third example of similar interactions is the recognition of the transcription factor sigma[32] by *E. coli* Hsp70 DnaK: The DnaK binding sites are located in a helix that became transiently unfolded enabling interaction with DnaK. Intriguingly, the DnaK co-chaperone DnaJ also binds to sigma[32], resulting in its destabilization near the DnaK binding site (51), supporting a possibility that a similar (co-chaperone dependent) mechanism can be realised for BiP-Ire1 interactions in the absence of substrate-induced Ire1 oligomerisation as reported by Amin-Wetzel and co-authors (18).

Growing evidence suggests that BiP regulates the Ire1 activation process using multiple dynamic mechanisms, enabling precise and stress-specific fine-tuning of the Ire1 conformational ensemble through highly specific interactions with different oligomeric species of Ire1-LD (Fig 5). Contributions of individual BiP binding modes in vivo are likely to depend on the population of individual oligomeric species of Ire1-LD among Ire1 paralogs and orthologs. Interestingly, the strength of dimerization is significantly larger for human Ire1α-LD (with Ka is ca. 0.5 μM in this study and 2.5 μM for a shorter Ire1-LD construct that lacks residues 391–449 (13)) than for yeast Ire1-LD (Ka is ca. 8.2 μM (14)), suggesting that BiP contribution in regulation of Ire1 activation in the presence and the absence of stress can significantly vary between different species. Another factor that orchestrates the complex, multistep Ire1-LD activation process is the accessibility of multivalent interaction motifs in different Ire1-LD conformations. In turn, the process might also be fine-tuned by the concentration of unfolded proteins in the ER and their specific features, such as binding affinity to Ire1-LD, BiP and BiP co-chaperones, providing additional control in vivo.

In summary, our study revealed an evolutionarily conserved mechanism by which the ER Hsp70 molecular chaperone BiP actively reshapes and controls the Ire1 clustering induced by the accumulation of unfolded proteins in the ER. We found that two hydrophobic motifs located at the dynamic oligomerisation interface of Ire1-LD play a critical role in BiP interactions. Uncovering these molecular details provides exciting opportunities in the modulation of the Ire1 activation process, UPR, and cell fate that could potentially lead to comprehensive mechanical understanding of UPR signalling in health and many UPR-associated diseases.

## Materials and Methods

### Expression and purification of Ire1-LD and BiP

The human Ire1α-LD (aa 24–449, called Ire1-LD, UniProt ID: O75460) was cloned from the full-length protein (a kind gift from David Ron, University of Cambridge) into a pET His6 TEV LIC cloning vector (a gift from Scott Gradia, plasmid #29653; Addgene; http://n2t.net/addgene:29653; RRID:Addgene_29653). We designed oligomerisation-deficient

constructs (D123P, [315]LL[316] to DA, [358]WLLI[362] to GSSG) and the truncated constructs (that comprise residues 24–390 and 24–356) using 2X High-Fidelity Q5 Polymerase Master Mix from New England Biolabs and primers purchased from Integrated DNA Technologies. WT BiP and its variants (BiP T229G and BiP V461T) that contain a non-cleavable N-terminal 6x-His-tag were cloned into the pET28a vector ([33]). Alternatively, BiP with a TEV-cleavable N-terminal 6x-His-tag (called BiP* in the text) was cloned into a pET-29b(+) cloning vector (Twist Bioscience). The protein sequences of expressed Ire1-LD and BiP variants are provided at ([52]).

The labelled and unlabelled proteins were expressed as described previously ([33]) in *E. coli* BL21 (DE3) competent cells. A single colony was resuspended in 2 ml LB broth (Thermo Fisher Scientific) supplemented with kanamycin and grown overnight at 37°C. The small amount of the overnight culture was transferred into 500 ml LB broth media to reach starting optical density at a wavelength of 600 nm ($OD_{600}$) ~0.1. The culture was incubated at 37°C with shaking until $OD_{600}$ reached ~0.8. Then protein expression was induced with IPTG (final concentration 1 mM), and the culture was grown for another 4–6 h at 37°C for all BiP and Ire1-LD constructs except the truncated Ire1-LD (residues 24–390 and 24–356), D123P and [358]WLLI[362] to GSSG Ire1-LD constructs. To produce samples for these constructs, the induced culture was left to grow overnight (ca. 15 h) at 20°C with shaking. Cells were harvested by centrifugation, resuspended in the binding buffer (20 mM HEPES, 400 mM NaCl, pH 8.0), and frozen at –80°C until the purification step.

Expression of $^2$H, $^{15}$N, Ile$\delta$1-[$^{13}$CH$_3$]-labelled BiP and $^{15}$N-labelled Ire1-LD samples were performed according to published method ([33]). Particularly, for $^2$H, $^{15}$N, Ile$\delta$1-[$^{13}$CH$_3$]-labelling ([53]), transformants were grown in 2 ml LB broth for 6 h. Cells were harvested and transferred to 25 ml M9 minimum media (starting $OD_{600}$ ~0.1) containing deuterated D-glucose (CIL, 1,2,3,4,5,6,6-D7, 98% DLM-2062), $^{15}$N-labelled ammonium chloride ($^{15}$NH$_4$Cl), 10% of deuterated Celtone Complete Medium (CIL, CGM-1040-D) and D$_2$O, and grew overnight at 30°C. The next day, the pre-culture was transferred to 500 ml labelled M9 media (starting $OD_{600}$ ~0.2) and incubated at 37°C. When the $OD_{600}$ reached ~0.7, 5 ml of methyl-$^{13}$C-labelled alpha-ketobutyric acid (CLM-6820) solution (14 mg/ml in D$_2$O, pH 10.0) was added. After ~1 h of incubation at 37°C, IPTG was added to the final concentration of 1 mM. The protein was expressed for 6–7 h before harvesting as described above. For $^{13}$C,$^{15}$N and $^{15}$N labelling, samples were produced using the same protocol as for unlabelled samples, except M9 minimum media containing 1 g/liter and 2 g/liter of $^{13}$C-labelled D-glucose and/or $^{15}$N-labelled ammonium chloride ($^{15}$NH$_4$Cl) were used instead of LB broth.

For all samples, thawed cells were incubated for 30 min on ice with 0.5 ml lysozyme (50 mg/ml; Sigma-Aldrich), 0.1 ml DNaseI (10 mg/ml; Sigma-Aldrich), and Protease Inhibitor Cocktail (Roche). After sonication, lysates were spun down at 22,000*g* for 45 min and filtered to remove cell debris. Supernatants were loaded onto a HisTrap HP nickel column (GE Healthcare) pre-equilibrated with the binding buffer (20 mM HEPES, 400 mM NaCl, pH 8.0). After a wash step with the binding buffer containing 40 mM imidazole, the bound protein was eluted by 500 mM imidazole. Collected fractions were extensively dialysed against the final HMK buffer (20 mM HEPES, 100 mM KCl, 5 mM MgCl$_2$, pH 7.6), flash frozen in liquid nitrogen, and

stored at –80°C. Protein concentrations were measured using UV absorbance at 280 nm. The purity of the samples was characterized using the SDS–PAGE gel and by measuring the A260/A280 ratio. In addition, protein concentrations were measured using the Bradford protein assay (Bio-Rad) with bovine serum albumin as a standard.

It has been demonstrated previously that the His-tag does not significantly affect BiP function ([29]). To test whether the His-tag affects substrate binding and BiP ability to de-oligomerize Ire1-LD (see below), WT BiP was incubated for 18 h at 4°C with the His-tagged TEV protease (the 1:10 protein:TEV ratio) in the presence of 0.5 mM EDTA and 5 mM DTT. The reaction was loaded onto a HisTrap HP nickel column (GE Healthcare) pre-equilibrated with the binding buffer (20 mM HEPES, 400 mM NaCl, pH 8.0) to remove His-tagged TEV and non-cleaved BiP.

### Peptides

The following peptides were used in this study: ΔEspP (sequence: MKKHKRILALCFLGLLQSSYSAAKKKK ([14])); MZP1 peptide (sequence: LIRYCWLRRQAALQRRISAME ([13])); NR peptide sequence (HTFPAVL ([33])); Ire1-derived peptides: AVVPRGS, GSTLPLLE, RNYWLLI, KHRENVI, ENVIPADS, and KDMATIIL. All peptides were purchased from BioMatik, resuspended in DMSO at a concentration of 100 mM or (when possible) in water at a final concentration of 2 or 10 mM.

### MST experiments and data analysis

To label Ire1-LD with FITC, 50 $\mu$M of 1 M sodium carbonate at pH 8.0 and 9.6 $\mu$l of 1 mg/ml DMSO solution of FITC were added to 500 $\mu$l of 25 $\mu$M of the protein sample and incubated for 60 min. To remove excess FITC, the reaction was loaded onto a NAP5 column pre-equilibrated with HMK. The protein concentration was measured using A280 and A495 and calculated using the following equation: A280$_{corrected}$ = A280 − 0.35 (A495). A dilution series (the FITC-labelled Ire1-LD concentrations range from 5 nM to 100 $\mu$M) was prepared in the HMK buffer in the presence of 1 mM TCEP or 5 mM DTT. The measurements were performed using a NanoTemper Monolith NT 1.15; data were analysed using NanoTemper Monolith NT 1.2.101 Analysis software. Dissociation constants reported were the average of three independent experiments. Error bars represent standard errors.

### SEC experiments and data analysis

All SEC was carried out using HMK buffer with 5 mM DTT added unless stated otherwise using a Superdex 200 10/300 column (GE healthcare Life Sciences). For each experiment, 300 $\mu$l of protein sample was injected into the column and passed through the column with a flow rate of 0.35 ml/min at 4°C, and the absorbance at 280 nm was recorded for eluting species. A calibration curve was produced for eluting species using known protein standards (GE Healthcare Life Sciences). The dimerization-impaired D123P construct of Ire1-LD was used to assign the apparent molecular weight of eluted monomeric protein, which was subsequently used to calculate oligomerisation state of Ire1-LD samples.

## Dynamic light scattering

If required, 10 $\mu$M $\Delta$EspP and/or 5 $\mu$M WT BiP and 10 mM ATP were added to A 5 $\mu$M Ire1-LD sample in HMK buffer; each sample was centrifuged at 16,000$g$ for 5 min and filtered through a 0.22 $\mu$M filter. A Wyatt miniDawnTreos system was used for the measurements. Buffer was injected, and equilibrium was reached over 5 min, then the sample was injected, and data were recorded for 3 min. Between measurements, the system was washed with 1 M nitric acid and d$H_2$O. Astra 6.0.3 software was used to analyse the data using the cumulants model (54).

## Flow-induced dispersion analysis (FIDA)

A 10 $\mu$M Ire1-LD sample in HMK buffer was centrifuged at 16,000$g$ for 5 min and then filtered through a 0.22 $\mu$M filter. 1 $\mu$M $\Delta$EspP was added to the protein and the samples were incubated for 10 min, after which 1 or 0.1 $\mu$M WT BiP and 40 mM ATP were added if required. The samples were pipetted into a glass vial with an insert or 96-well plate. Buffer was flushed through the capillary at 3,500 mbar for 20 s, then by injection of 40 nl of the protein sample at 50 mbar for 10 s. The sample plug was then pushed through with buffer at 400 mbar for 240 s to give the Taylorgram (the measurements of intrinsic fluorescence intensity for each data point). The measurements were performed at 25°C using the Fida Biosystem software v2.44 (Fida Biosystems ApS). The apparent protein hydrodynamic radius for each condition was obtained using the Fida Biosystems software v2.34 (Fida Biosystems ApS) as previously described (55).

## Electron microscopy (EM)

For electron microscopy (EM) the luminal domain at a final concentration of 20 $\mu$M was incubated with 75 $\mu$M of $\Delta$EspP for 2 h and diluted 10-fold. The sample was then negative stained: 3 $\mu$l of the diluted reaction was taken and added to a glow-discharged carbon-coated copper grid (produced at the University of Leeds Astbury BioStructure Laboratory) for 30 s before being washed with 5 $\mu$l of d$H_2$O for 3 s, with excess liquid being blotted away in between steps. 5 $\mu$l of 2% uranyl acetate was then added to the grid and left for 3 s; this step was repeated once more. Lastly, 5 $\mu$l of 2% uranyl acetate was added to the grid again, but for 30 s, this time before being blotted away extensively. Images were taken in the University of Leeds Astbury BioStructure Laboratory using a FEI T12 microscope with a Lab6 filament and Gatan UltraScan 4000 CCD camera.

## Ire1-LD solubility assay

To determine the concentration of soluble Ire1-LD, the reaction mixture (20 $\mu$M of Ire1-LD and the required concentration of $\Delta$EspP in HMK buffer) was incubated for 2–3 h at room temperature. The reaction was then centrifuged at 16,000$g$ for 5 min and the Ire1-LD concentrations in supernatant and pellet (after washing three times in HMK buffer) were measured using the Bradford protein assay (Bio-Rad) with bovine serum albumin as a standard. SDS–PAGE gel samples were also taken of the supernatant and the pellet. Complementarily, aliquots of soluble fractions were taken to

measure protein concentrations using the Bradford assay. The amounts of soluble protein obtained from the SDS–PAGE analysis and solubility assay were in good agreement for all measurements. If required, the different amount of BiP was added to the reaction mixture and after 3 h of incubation with BiP; 40 mM of ATP was added to the reaction and incubated for another 3 h. The 40 mM ATP was added for consistency with other experiments and to ensure that the reaction mixture contains a constant concentration of ATP during the long incubation period; however, 1 mM ATP was sufficient for a shorter incubation period (Fig S17A and B).

## Ire1-LD turbidity assay

The desired concentration of $\Delta$EspP or MZP1 was added to 20 $\mu$M of Ire1-LD (or its variants) in HMK buffer in the presence of 1 mM TCEP; BiP (10, 2, or 0.2 $\mu$M) and ATP (40 mM) were added if required. The reaction was incubated for 5 min at 25°C or 30°C. Ire1-LD and BiP protein stocks were centrifuged at 12,000$g$ for 15 min to remove all insoluble material. The absorbance at 400 nm (OD400) was measured as described previously (56) using a POLARstar OPTIMA microplate reader (BMG Labtech Ltd). The measurements were taken every 2, 3, or 10 min after 5 min of an initial incubation. For each condition, reactions were prepared in triplicates.

## ATPase measurements

ATPase activity was measured as previously described (33) using an ATPase/GTPase activity assay kit (MAK113; Sigma-Aldrich) according to the manufacturer's protocol. The amount of released inorganic phosphate ($\mu$moles Pi per hour per moles BiP) was detected to measure ATPase activity. Briefly, samples were prepared by addition of the BiP* and peptide substrates dissolved in DMSO (up to 1% DMSO in total) to a total volume of 30 $\mu$l in each well; 10 $\mu$l of 4 mM ATP was added to start the reaction. The final concentrations were 1 mM of ATP, 1 $\mu$M of BiP*, and 1 mM of peptide substrate in HMK buffer. Samples were incubated at 37°C for 1 h, and then 200 $\mu$l of the manufacturer's reagent was added to each well. Samples were mixed and incubated at room temperature for 30 min, and the OD$_{620}$ was measured. Measurements were performed in triplicate.

## SDS–PAGE gel analysis of protein expression

To elucidate whether the Ire1-LD variants can be expressed in *E. coli*, the cell pellet was resuspended in 250 $\mu$l of BugBuster Master Mix (Millipore) and incubated at room temperature for 20 min. After centrifugation at 16,000$g$, to prepare the soluble protein fraction, 20 $\mu$l of the supernatant was taken and added to 8 $\mu$l 4X Laemmli sample buffer (Bio-Rad) and 12 $\mu$l of 8 M urea. To prepare the insoluble fraction, the pellet was washed three times with LB media before being resuspended in 16 $\mu$l of 4X Laemmli sample buffer and 24 $\mu$l of 8 M urea.

## NMR experiments

### Solution NMR

To obtain fingerprints of the conformational states of the BiP functional cycle in the presence and absence of unfolded protein

substrates, the ATPase deficient BiP* T229G variant (29) was expressed in isotopically labelled media as described above. NMR acquisitions were carried out in the HMK buffer (20 mM HEPES, 100 mM KCl, 5 mM MgCl$_2$, pH 7.6) with 40 mM of ATP if required. A band-selective optimised-flip-angle short-transient experiment (57) (2D $^1$H-$^{13}$C SOFAST-HMQC) was used to acquire 2D methyl NMR spectra (58). All measurements were recorded at 25°C on 750 MHz or 950 MHz Bruker spectrometers equipped with a 5 mm (750 MHz) or 3 mm (950 MHz) Bruker TCI triple-resonance cryogenically cooled probe. Data were processed with NMRPipe (59) and analysed with CcpNmr Analysis software packages (60, 61). Peaks corresponded to domain-docked and -undocked conformations of BiP were assigned as described previously (33). As has been previously demonstrated, substrate binding affects the equilibrium of the BiP conformational ensemble, favouring the domain-docking confor-mation (33). The presence of the His-tag does not perturb substrate binding, as monitored by methyl NMR. To calculate the populations of the domain-docked and -undocked conformations from spectra of BiP* T229G, we used methyl peak intensities of three non-overlapping peak doublets (same as in (33)). Individual peaks in each doublet correspond to the domain-docked and -undocked conformations. Peak intensities were obtained using the parabolic method implemented in CcpNmr (60, 61). The population of the domain-docking conformation was calculated as $p_D = I_D/(I_D + I_U) \times 100\%$. Errors were calculated as standard deviations (SDs) from the means for three peak doublets.

To elucidate Ire1-LD conformational features of Ire1-LD, $^{15}$N-labelled Ire1-LD and its truncated constructs (containing residues 24–390 and 24–356) were expressed in isotopically labelled media as described previously. NMR acquisitions were performed in the HMK buffer. A 2D amide BEST TROSY correlation experiment was used to acquire 2D amide NMR spectra; the final concentration of Ire1-LD was 50 $\mu$M. All measurements were recorded on 750 MHz and 950 MHz Bruker spectrometers equipped with a 5 mm (750 MHz) or 3 mm (950 MHz) Bruker TCI triple-resonance cryogenically cooled probe. To extract amide proton temperature gradients, 2D spectra were recorded at 5°C, 10°C, 20°C, 25°C, and 30°C. The proton chemical shifts of each peak were then plotted as a function of temperature and fitted linearly to obtain temperature gradient values, as previously described (38). Data were processed with NMRPipe (59) and analysed with CcpNmr Analysis software packages (60, 61).

### Solid-state NMR

205 $\mu$M of ΔEspP peptide was added to 40 $\mu$M $^{13}$C,$^{15}$N Ire1-LD in HMK buffer containing 5 mM DTT; the sample then was incubated at room temperature for 3 h. The sample was left overnight at 4°C, then centrifuged at 16,000$g$ for 5 min to collect the pellet. The experiments were carried out at 60 kHz magic angle spinning frequency on a Bruker Avance HD spectrometer at 700.06 MHz $^1$H Larmor frequency using 1.3 mm HXY probe in triple-resonance mode. The sample was cooled with a gas flow of 700 liters/h (0°C and –27°C on the sensors at the top and bottom of the probe), which resulted in the effective sample temperature of ~26°C as estimated from the difference between the bulk water and DSS peaks. Chemical shifts were internally referenced with respect to $^1$H peak of DSS using IUPAC recommendations for indirect referencing. $^1$H-detected $^1$H-$^{13}$C and $^1$H-$^{15}$N 2D correlation spectra were recorded using both

cross-polarisation (CP) and INEPT-based $^1$H-$^{13}$C/$^{15}$N polarisation transfer. The lengths of CPs were: 300/300 $\mu$s ($^1$H-$^{13}$C CP/$^{13}$C-$^1$H CP) and 700/400 $\mu$s ($^1$H-$^{15}$N CP/$^{15}$N-$^1$H CP) to produce mostly one-bond correlations. The INEPT delays were optimised for maximum signal resulting in delays of 0.9/1.2 ms for $^1$H-$^{13}$C experiment and 1.6/1.8 ms for $^1$H-$^{15}$N experiment. The MISSISSIPPI solvent suppression scheme was applied with a spinlock field of ~50 kHz for four 80 ms in all experiments. Recycle delay of 2 s was used for all experiments. $^1$H-$^{13}$C CP-based experiment was acquired with 80 transients, $^1$H-$^{13}$C INEPT-based experiment was acquired with 192 transients, $^1$H-$^{15}$N CP-based experiment was acquired with 64 transients, $^1$H-$^{15}$N INEPT-based experiment was acquired with 1,024 transients.

### Liquid chromatography-mass spectrometry

The mass analysis was performed by LC-MS using an M-class ACQUITY UPLC (Waters UK) interfaced to a Synapt G2S Q-IMT-TOF mass spectrometer (Waters UK). 1 $\mu$l of 5 $\mu$M Ire-LD sample was loaded onto a MassPREP protein desalting column (Waters UK). The column eluent was directed into the mass spectrometer via a Z-spray electrospray source. Mass calibration was performed by a separate injection of [Glu]-fibrinopeptide b at a concentration of 250 fmol $\mu$l$^{-1}$ in MS/MS mode and a CID voltage (trap region) of 28 V. Data processing was performed using the MassLynx v4.1 suite of software (Waters UK) supplied with the mass spectrometer.

### Sequence conservation analysis

The Consurf server (35, 36) was carried out using default param-eters, except a minimal percentage identity between homologue of 30%. The human ERN1 sequence (UniProt ID O75460) was used as an input sequence. 124 HMMER-identified homologues were selected automatically to produce conservation scores and phylogenetic tree. The phylogenetic tree was visualized in the interactive Tree of life tool (iTol) (37). The model 3D structures were obtained from AlfaFold2 (62) (Sept 2023) and analysed using Molecular Graphics System, Version 2.5.4 Schrödinger, LLC.

## Data Availability

The data from this publication have been deposited to the Uni-versity of Leeds database at (52).

## Supplementary Information

## Acknowledgements

This work was supported by BBSRC grant BB/M021874/1. We thank Dr Geoff Kelly (MRC Biomedical NMR Centre, The Francis Crick Institute, London, UK) for assistance with NMR data collection; Dr James Ault and Dr Rachel George (University of Leeds Mass Spectrometry Facility) for mass spectrometry data

collection and analysis; Dr Rebeca Thompson (University of Leeds cryo-Electron Microscopy facility). We acknowledge the Astbury Biostructure Laboratory for access to the 750 and 950 MHz NMR spectrometer which was funded by the University of Leeds and the Francis Crick Institute through the provision of access to the Medical Research Council Biomedical NMR Centre. The Francis Crick Institute receives its core funding from Cancer Research UK (FC001029), the Medical Research Council (FC001029) and the Wellcome Trust (FC001029). We acknowledge Wellcome for funding the Monolith NT.115 MST (105615/Z/14/Z) and ITC instruments (094232/Z/10/Z). We are grateful to Fidabio for the loan of a Fida 1 machine. For the purpose of Open Access, the authors have applied a CC BY public copyright licence to any Author Accepted Manuscript version arising from this submission.

## Author Contributions

S Dawes: data curation, formal analysis, validation, investigation, visualization, methodology, and writing—original draft, review, and editing.

N Hurst: data curation, formal analysis, validation, investigation, visualization, methodology, and writing—original draft, review, and editing.

G Grey: investigation, visualization, and writing—review and editing.

L Wieteska: validation, investigation, visualization, and writing—review and editing.

NV Wright: investigation, visualization, and writing—review and editing.

IW Manfield: supervision, investigation, methodology, and writing—review and editing.

MH Hussain: investigation and methodology.

AP Kalverda: supervision, investigation, methodology, and writing—review and editing.

JR Lewandowski: data curation, validation, investigation, methodology, and writing—review and editing.

B Chen: conceptualization, supervision, funding acquisition, validation, and writing—review and editing.

A Zhuravleva: conceptualization, data curation, formal analysis, supervision, funding acquisition, validation, investigation, visualization, methodology, project administration, and writing—original draft, review, and editing.

## Conflict of Interest Statement

The authors declare that they have no conflict of interest.

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
