## [Reviewer comments · Life Science Alliance]

Life Science Alliance

Chaperone BiP controls ER stress sensor Ire1 through interactions with its oligomers

Sam Dawes, Nicholas Hurst, Gabriel Grey, Lukasz Wieteska, Nathan Wright, Iain Manfield, Mohammed Hussain, Arnout Kalverda, Jozef Lewandowski, Beining Chen, and Anastasia Zhuravleva

DOI: <https://doi.org/10.26508/lsa.202402702>

Corresponding author(s): Anastasia Zhuravleva, University of Leeds

Review Timeline:

Submission Date:	2024-03-08
Editorial Decision:	2024-04-10
Revision Received:	2024-07-15
Editorial Decision:	2024-07-16
Revision Received:	2024-07-24
Accepted:	2024-07-24

Transaction Report:

April 10, 2024

Re: Life Science Alliance manuscript #LSA-2024-02702-T

Anastasia Zhuravleva
University of Leeds
United Kingdom

Dear Dr. Zhuravleva,

Thank you for submitting your manuscript entitled "Chaperone BiP controls ER stress sensor Ire1 through interactions with its oligomers" to Life Science Alliance. The manuscript was assessed by expert reviewers, whose comments are appended to this letter. We invite you to submit a revised manuscript addressing the Reviewer comments.

Thank you for this interesting contribution to Life Science Alliance. We are looking forward to receiving your revised manuscript.

Sincerely,

B. MANUSCRIPT ORGANIZATION AND FORMATTING:

Reviewer #1 (Comments to the Authors (Required)):

15-Mar-2024

This paper takes on the unresolved question of the molecular basis for activation of the Unfolded Protein Response transducer IRE1 by endoplasmic reticulum stress. This is a topic of fundamental importance to life science and the advances reported here are worthy of our attention and critical assessment.

Working in vitro and using isolated proteins and peptides the authors make several observations that are then linked to a hypothetical framework for understanding an aspect of IRE1 regulation.

Using a variety of biophysical techniques (SEC, DLS, TEM imaging) they report on the ability of a synthetic peptide, Δ EspP, to promote high order insoluble oligomers of the stress-sensing luminal domain of human IRE1a. This process is disrupted by mutation in conserved residues in IRE1a putative homo-oligomerization surfaces. Furthermore, the IRE1 molecules incorporated into the oligomers are reported to retain their folded state. The authors suggest that this experimental setup mimics an activation step that normally occurs in cells.

Unlike soluble IRE1, these IRE1 oligomers attract the ER chaperone BiP. Measurable BiP binding occurs in the chaperone's apo state, however, in the presence of ATP, BiP solubilises IRE1, extracting it from the insoluble oligomers induced by the Δ EspP peptide. This feature is very dramatic and observed in BiP:IRE1 stoichiometry as low as 1:100. Furthermore, BiP compromised either in its ability to bind substrates (V461F) or in its ability to hydrolyse ATP (T229G) is ineffective in this solubilisation assay. This experiment implies that BiP is acting catalytically on the insoluble oligomers, via cycles of ATP hydrolysis and nucleotide exchange. It is notable that this in vitro activity proceeds with neither a J-domain protein to stimulate ATP hydrolysis or nucleotide exchange factor to facilitate BiP re-cycling by exchange of ADP for ATP. The authors suggest that this experimental set up is an in vitro counterpart to BiP mediated repression of IRE1 activity.

Algorithms for BiP binding predict it to engage two peptides in the flexible C-terminal region of IRE1a luminal domain. These peptides do indeed bind to isolated BiP as reflected by their ability to perturb its NMR spectrum and to enhance ATP hydrolysis. Lastly the authors use Alpha Fold to predict the disposition of these BiP binding peptides, in various configurations of IRE1. Based on these last two observations the authors propose that activating ligands, modelled here by Δ EspP, promote IRE1 oligomerisation to expose these BiP binding peptides from IRE1's C-term. In the absence of ATP, BiP engages the peptides stably and is itself incorporated into the insoluble oligomers, but in presence of ATP, BiP can cycle and 'work' on the oligomers undoing them, a counterpart to BiP repression of IRE1 activity.

Critique:

Even if we accept the author's claim that Δ EspP-mediated oligomerisation observed here is relevant to IRE1 activation in cells, and if we accept that BiP binding occurs through the IRE1 peptides identified here, it remains unproven that the de-oligomerisation process is mediated through such binding, though this remains a plausible idea.

The notion that IRE1 assembly into large clusters is a prerequisite to its activation has received something of rude awakening when the Walter lab -for many years proponents of the idea - reported that single particle tracking of IRE1 expressed at endogenous levels, failed to support the existence of the large oligomers suggested previously by over-expression studies. Rather, activation entailed at most a transition from dimers to small oligomers, likely tetramers (Belyy et al., 2022). It remains possible that the insoluble oligomers described here are an exaggerated form of the small oligomers shown by Belyy and colleagues to reflect IRE1's activate state and that BiP repression of IRE1 entails their undoing. It is also possible, however that the binding events described here are a counterpart to the observations of Amin-Wetzel and colleagues who mapped the region on IRE1 necessary for BiP mediated repression to the flexible C-term of its luminal domain (Amin-Wetzel et al., 2019), a region that contains the BiP-binding peptides identified here. This interpretation detracts in no way from the importance of the author's findings and aligns them better with current views on IRE1 function, views that challenge the importance of very large assemblies of a molecule expressed at very low levels in cells. The authors are encouraged to reflect on these points as they revise their discussion.

The Δ EspP-mediated oligomerisation of the human IRE1a luminal domain observed here is at odds with the observations of Carrara and colleagues: Using SEC, they noted no effect of Δ EspP, when added at 500 μ M to the human IRE1a luminal domain (see panel D, supplement 1 to Figure 4 in their paper)(Carrara et al., 2015). Indeed, it is surprising that Δ EspP, a peptide proposed to bind the distantly related yeast IRE1 luminal domain, would be a better activating ligand than the MPZ peptide, identified by Karagoz and colleagues as a ligand of human IRE1a (Karagoz et al., 2017), the molecule studied here by Dawes and colleagues. This reviewer would urge caution when it comes to the question of IRE1 activating ligands: Competable binding of such ligands to an IRE1a luminal domain has not been reported and the binding site(s) have never been mapped adequately (the notion that binding occurs in the groove between IRE1a protomers, is particularly poorly supported). It remains possible that the entire literature on IRE1 activating ligands is based on irrelevant in vitro artefacts.

40 mM ATP is added to promote BiP-mediated solubilisation of IRE1 oligomers (Figure 2 panel d), why is such a high

concentration needed?

For BiP to act catalytically in the disassembly process, the kinetics of BiP binding and release and the kinetics of re-oligomerisation of the released IRE1 need to be somehow matched (for example, the latter needs to be slow). The paper would have been elevated to higher level if these parameters were measured experimentally. But even without doing so, a discussion would benefit the readers.

References

- Amin-Wetzel, N., Neidhardt, L., Yan, Y., Mayer, M.P., and Ron, D. (2019). Unstructured regions in IRE1 α specify BiP-mediated destabilisation of the luminal domain dimer and repression of the UPR. *eLife* 8, e50793.
- Belyy, V., Zuazo-Gaztelu, I., Alamban, A., Ashkenazi, A., and Walter, P. (2022). Endoplasmic reticulum stress activates human IRE1 α through reversible assembly of inactive dimers into small oligomers. *eLife* 11.
- Carrara, M., Prisci, F., Nowak, P.R., Kopp, M.C., and Ali, M.M. (2015). Noncanonical binding of BiP ATPase domain to Ire1 and Perk is dissociated by unfolded protein CH1 to initiate ER stress signaling. *eLife* 4.
- Karagoz, G.E., Acosta-Alvear, D., Nguyen, H.T., Lee, C.P., Chu, F., and Walter, P. (2017). An unfolded protein-induced conformational switch activates mammalian IRE1. *eLife* 6.

Reviewer #2 (Comments to the Authors (Required)):

Summary:

ER stress plays critical roles in physiological and pathological events. The inositol-requiring enzyme 1 (IRE1) is one of the three widely recognized sensors of unfolded protein response (UPR). In this manuscript, Dawes et al., demonstrated that Bip regulated Ire1 by interacting in a ATP dependent manner. Furthermore, they identified 2 binding motifs of Ire1 interacting with Ire1. However, several significant concerns have been raised by reviewers:

Major Concerns:

1. This paper used a "clean" model consisting of proteins and peptides to study the interactions, but the interaction in cultured cells or in vivo will be more complicated. Some of the key findings should be evaluated in such a more physiological context.
2. A concern raised from Fig.4b the most conserved scores including 306VVP308 are not within the 2 binding motif including 310GSTLPLL316.
3. Fig2b left panel, To prove it's ATP depend. there should have control with no ATP but Δ EspP+different ratio of Bip. In the middle panel, the 2nd and 3rd lane indicating T229G and V461F mutations inactivate BiP, please explain how you inactivated the BiP for 1st.
4. There should be a loading control for WB in Fig. 2c and 2d. Showing the ratio of soluble and insoluble protein would be more straight to help support the conclusion.
5. Please add a list of abbreviations.
6. Please add the statistic analysis statement in the methods

Thoughtfully addressing these major and minor concerns will undoubtedly strengthen this review and enhance its potential impact in the field.

Reviewer #3 (Comments to the Authors (Required)):

In this study the researchers repeated the oligomerization of Ire1-LD induced by two reported model peptides, and based on this they found the size of Ire1-LD oligomers depends on peptide concentration and/or substrate affinity. The interesting finding is that insoluble Ire1-LD. Δ EspP oligomers have fibril-like morphology and folded-protein-like structure. Then the researchers found that Bip de-oligomerized the Ire1-LD. Δ EspP oligomers in an ATP-dependent way by interacting with two motifs in this flexible C-terminal region of Ire1-LD. Both motifs are near to the oligomerization interface of Ire1-LD and the region containing the two motifs are highly dynamics. So this study contributes some interesting and new findings to the community. However related deep digging is still not enough for these new findings. Some detailed considerations are as the following.

1. The structural feature of Ire1-LD was not described clearly, and in the structure of Ire1-LD. Δ EspP oligomers how the monomers arrayed is also not clear. Is there some evidence which showed that Δ EspP was in the oligomers? It is still a mystery how Δ EspP interacts with Ire1-LD, and then induces conformational change of Ire1-LD and its oligomerization. Although there is some conformational change of Ire1-LD upon binding of Δ EspP the researches still found Ire1-LD oligomer adopted its folded conformation. The binding ratio and mode between Ire1-LD and Δ EspP is unclear. The binding site in Ire1-LD of Δ EspP or the unfolded protein is also unknow. Do Δ EspP and the unfolded protein possibly share the same binding site in Ire1-LD? What is the difference between the fibrillar structure of Ire1-LD. Δ EspP oligomers and amyloid fibrillar structure?

2. It is reasonable that Bip cannot form stable interaction with Ire1-LD in the presence of ATP since ATP-state Hsp70 has lower affinity with substrate. In the presence of ADP or in the absence nucleotide, Hsp70 has higher affinity with substrate. However stable interaction between Bip and Ire1-LD is also not observed in the absence of ATP. In supplementary Figure 10 Bip was added into the labeled Ire1-LD oligomers to check their interaction. If the binding site in Ire1-LD is highly dynamics the NMR

signal of this region will be very weak, and the signal change will not be so easy to be detected. However Ire1-LD dimer or oligomers were not added into labeled Bip to test the interaction as done for the two short peptides. It is interesting to find two Bip-binding peptides in the C-terminal region of Ire1-LD. However the affinity of their binding to Bip was not measured. The substrate promoted ATPase activity was only measured in one ratio of peptide: Bip (1:1?). It is better to observe elevated promotion with higher ratio of peptide: Bip. The effect of Bip binding to Ire1-LD. Δ EspP oligomers on the interaction between Ire1-LD and Δ EspP is unknown. Is there competition between binding of Bip and Δ EspP to Ire1-LD? Although it is found that the oligomerization interface and BiP binding sites of Ire1-LD is highly dynamic, it is unknown whether the dynamics changes with BiP binding, Δ EspP binding and Ire1-LD oligomerization.

3. In the method about ATPase measurements, 1 mM of Hsp70 for ATPase assay is unusual. After malachite green reagent was added, 34% sodium citrate were added normally.

4. In the method about protein purification, Hamster and human BiP were purified. However it unclear when they are used in different experiments. Although His tag will not affect ATPase activity it could affect substrate binding. So it should be careful if His tag was not removed in some experiments.

5. Some small format problems, such as no blank between number and unit in some places (the last paragraph in P15) , missing subscript (H₂O in P15).

Reviewer #1 (Comments to the Authors (Required)):

15-Mar-2024

This paper takes on the unresolved question of the molecular basis for activation of the Unfolded Protein Response transducer IRE1 by endoplasmic reticulum stress. This is a topic of fundamental importance to life science and the advances reported here are worthy of our attention and critical assessment.

Working in vitro and using isolated proteins and peptides the authors make several observations that are then linked to a hypothetical framework for understanding an aspect of IRE1 regulation.

Using a variety of biophysical techniques (SEC, DLS, TEM imaging) they report on the ability of a synthetic peptide, Δ EspP, to promote high order insoluble oligomers of the stress-sensing luminal domain of human IRE1a. This process is disrupted by mutation in conserved residues in IRE1a putative homo-oligomerization surfaces. Furthermore, the IRE1 molecules incorporated into the oligomers are reported to retain their folded state. The authors suggest that this experimental setup mimics an activation step that normally occurs in cells.

Unlike soluble IRE1, these IRE1 oligomers attract the ER chaperone BiP. Measurable BiP binding occurs in the chaperone's apo state, however, in the presence of ATP, BiP solubilises IRE1, extracting it from the insoluble oligomers induced by the Δ EspP peptide. This feature is very dramatic and observed in BiP:IRE1 stoichiometry as low as 1:100.

Furthermore, BiP compromised either in its ability to bind substrates (V461F) or in its ability to hydrolyse ATP (T229G) is ineffective in this solubilisation assay. This experiment implies that BiP is acting catalytically on the insoluble oligomers, via cycles of ATP hydrolysis and nucleotide exchange. It is notable that this in vitro activity proceeds with neither a J-domain protein to stimulate ATP hydrolysis or nucleotide exchange factor to facilitate BiP re-cycling by exchange of ADP for ATP. The authors suggest that this experimental set up is an in vitro counterpart to BiP mediated repression of IRE1 activity.

Algorithms for BiP binding predict it to engage two peptides in the flexible C-terminal region of IRE1a luminal domain. These peptides do indeed bind to isolated BiP as reflected by their ability to perturb its NMR spectrum and to enhance ATP hydrolysis.

Lastly the authors use Alpha Fold to predict the disposition of these BiP binding peptides, in various configurations of IRE1.

Based on these last two observations the authors propose that activating ligands, modelled here by Δ EspP, promote IRE1 oligomerisation to expose these BiP binding peptides from IRE1's C-term. In the absence of ATP, BiP engages the peptides stably and is itself incorporated into the insoluble oligomers, but in presence of ATP, BiP can cycle and 'work' on the oligomers undoing them, a counterpart to BiP repression of IRE1 activity.

Critique:

Even if we accept the author's claim that Δ EspP-mediated oligomerisation observed here is relevant to IRE1 activation in cells, and if we accept that BiP binding occurs through the IRE1 peptides identified here, it remains unproven that the de-oligomerisation process is mediated through such binding, though this remains a plausible idea.

The notion that IRE1 assembly into large clusters is a prerequisite to its activation has received something of rude awakening when the Walter lab -for many years proponents of the idea - reported that single particle tracking of IRE1 expressed at endogenous levels, failed to support the existence of the large oligomers suggested previously by over-expression studies. Rather, activation entailed at most a transition from dimers to small oligomers, likely tetramers (Belyy et al., 2022). It remains possible that the insoluble oligomers described here are an exaggerated form of the small oligomers shown by Belyy and colleagues to reflect IRE1's activate state and that BiP repression of IRE1 entails their undoing.

It is also possible, however that the binding events described here are a counterpart to the observations of Amin-Wetzel and colleagues who mapped the region on IRE1 necessary for BiP mediated repression to the flexible C-term of its luminal domain (Amin-Wetzel et al.,

2019), a region that contains the BiP-binding peptides identified here. This interpretation detracts in no way from the importance of the author's findings and aligns them better with current views on IRE1 function, views that challenge the importance of very large assemblies of a molecule expressed at very low levels in cells. The authors are encouraged to reflect on these points as they revise their discussion.

Many thanks for the very detailed and clear summary of the work and highlighting that the field remains highly controversial due to the complexity of the system. Indeed, while it has been shown that perturbations of Ire1-LD oligomerization significantly affect Ire1 activity *in vivo* (Ref 13), the role of small and large Ire1-LD oligomers in the oligomerization process yet remains elusive (Ref 8); the molecular details on Ire1-LD oligomerization have also remained unclear. Interestingly, at least *in vitro*, in the absence of protein substrate, Ire1-LD adopts predominantly monomeric and dimeric forms even at concentrations several magnitudes higher than its physiological concentrations, suggesting that oligomers' formation should be triggered by other (stress-related) factors. It has been also demonstrated by Walter's lab and others that unfolded protein substrates result in the formation of Ire1-LD oligomers *in vitro*, suggesting that similar interactions are likely to play an important role *in vivo* as well, providing a plausible mechanism of how accumulation of unfolded protein in the ER can directly trigger Ire1-LD activation. In this study, we further demonstrated that (at least *in vitro*) BiP plays a critical role on controlling Ire1-LD oligomerization and thus Ire1 activity.

We would like to highlight that BiP results in de-oligomerization of both insoluble Ire1-LD and soluble oligomers and larger insoluble oligomers. We have now added statements in the main text to highlight this. The key difference between Amin-Wetzel et al., 2019 and our studies is that BiP require assistance of its co-chaperone to interact with apo Ire1-LD but can directly interact with Ire1-LD oligomers in the presence of Δ EspP, suggesting that Δ EspP binding results in conformational rearrangements in Ire1-LD. It is, however, possible that BiP binds to the same Ire1-LD sites in both cases (as both sites are located in the C-terminal region of Ire1-LD). We have now added statements to the main text to highlight such a possibility. Alternatively, we cannot exclude the possibility that BiP could have different binding sites as these interactions control two different steps of Ire1-LD activation: oligomerization of Ire1-LD in the presence of substrate (in this study) and its dimerization in the absence of substrate (Amin-Wetzel et al., 2019). However, the detailed characterization of BiP-Ire1 interaction in the absence of substrate is a large undertaking and will be the focus of our future work.

The Δ EspP-mediated oligomerisation of the human IRE1a luminal domain observed here is at odds with the observations of Carrara and colleagues: Using SEC, they noted no effect of Δ EspP, when added at 500 μ M to the human IRE1a luminal domain (see panel D, supplement 1 to Figure 4 in their paper <https://elifesciences.org/articles/03522/figures#fig4s1> >)(Carrara et al., 2015).

We are not sure why the formation of high-order oligomers was not observed by Carrara et al., 2015; it could be, for example, due to a shorter incubation time with the peptide; for example, after a one-hour incubation of 20 μ M Ire1-LD with 200 μ M Δ EspP, ca. 50% of Ire1-LD is still soluble, while after two hours, more than 90% Ire1-LD is insoluble (see also Figure A below). We consistently observed the formation of soluble Ire1-LD oligomers in the presence of the peptides using a variety of experimental techniques, including DLS (dynamic light scattering) and FIDA (fluorescence intensity distribution analysis). In our experiments, SEC (size exclusion chromatography) at higher Ire1-LD and peptide concentrations was unreliable due to the significant formation of an insoluble Ire1-LD fraction at these concentrations after 0.5-2 hours of incubation with the peptide. For these reasons, in this

work we used significantly lower concentrations to ensure that Ire1-LD oligomers are soluble during the experiments (Figure 2a and Supplementary figures 9). Complementarily, we monitored the formation of insoluble oligomers (Figure 2c), which are, as we highlighted above, an extended version of the soluble Ire1-LD oligomers (see also Supplementary Figure 8) that form *in vitro* at higher Ire1-LD and peptide concentrations. Overall, we found that Ire1-LD oligomerization is a dynamic and complex process that depends on both the incubation time and the concentrations of Ire1-LD and peptide.

Indeed, it is surprising that Δ EspP, a peptide proposed to bind the distantly related yeast IRE1 luminal domain, would be a better activating ligand than the MPZ peptide, identified by Karagoz and colleagues as a ligand of human IRE1a (Karagoz et al., 2017), the molecule studied here by Dawes and colleagues.

We would like to highlight here that, similar to other PQC proteins, to detect ER stress (accumulation of unfolded proteins in the ER), Ire1-LD should be able to bind to a wide range of proteins rather than have a specific substrate. It has been shown previously that many hydrophobic sequences can potentially bind to yeast (Ref 14) and human Ire1-LD (Ref 13); the affinity of this binding is likely to depend on specific a.a. sequences and conformational features of these protein substrates, which are still to be uncovered. Interestingly, our study demonstrated that the strength of interactions between Ire1-LD and these model substrates is proportional to Ire1-LD oligomerization (Supplementary Figures 4 and 6) further supporting the specific nature of peptide binding and its importance for the Ire1-LD oligomerization process.

This reviewer would urge caution when it comes to the question of IRE1 activating ligands: Competable binding of such ligands to an IRE1a luminal domain has not been reported and the binding site(s) have never been mapped adequately (the notion that binding occurs in the groove between IRE1a protomers, is particularly poorly supported). It remains possible that the entire literature on IRE1 activating ligands is based on irrelevant *in vitro* artefacts.

We agree with the reviewer that the location of the binding site is a debatable question, but we are not trying to address this question in the current study, and it will be the focus of our future work. While it's quite plausible that the substrate binds to the binding cleft proposed for the structurally similar yeast Ire1-LD, we cannot, of course, fully exclude other possibilities and alternative binding site(s). However, even if the substrate binds to other parts of Ire1-LD, *in vitro* data suggest that interactions with the model peptide(s) control the formation of active (oligomeric) forms of Ire1-LD and, thus, cannot be ignored, especially considering the multicomponent and dynamic nature of the Ire1 activation process *in vivo*. Thus, while the field is still remained highly controversial, we believe that our work provides a better understanding of this complex system and the role of BiP in this process.

40 mM ATP is added to promote BiP-mediated solubilisation of IRE1 oligomers (Figure 2 panel d), why is such a high concentration needed?

The high concentration of ATP was chosen to maintain steady-state conditions during the incubation period: at this ATP concentration the ratio between BiP-ATP and BiP-ADP remains constant for approximately 12 hours as monitored by the relative peak intensities in the methyl NMR spectrum of methyl labelled BiP. Lower ATP concentrations could lead to ATP depletion during long experiments and, thus, complicate data interpretation. To address, the reviewer's concerns whether the BiP can only control Ire1-LD oligomerization

only at high (40 mM) ATP concentrations, now we repeated the experiments shown in Figure 2c with lower ATP concentrations (1 mM and 10 mM). To ensure that the ATP concentration is constant during the experiment, we used a shorter BiP incubation time (30 min instead of 2 hours). The results shown in the new Supplementary Figure 17, confirm that the presence of 1 mM ATP is already sufficient for BiP to de-oligomerize Ire1-LD.

For BiP to act catalytically in the disassembly process, the kinetics of BiP binding and release and the kinetics of re-oligomerisation of the released IRE1 need to be somehow matched (for example, the latter needs to be slow). The paper would have been elevated to higher level if these parameters were measured experimentally. But even without doing so, a discussion would benefit the readers.

The 'bulk' BiP-induced de-oligomerization of Ire1-LD is significantly faster (minutes vs. hours) than Ire1-LD oligomerization (Figure A below) with even a small amount of BiP (1:100 ratio) is already sufficient to de-oligomerize Ire1-LD and prevent the formation of larger oligomers. Nevertheless, the bulk measurements are difficult to interpret unambiguously due to complexity and heterogeneity of the system. Indeed, the Ire1-LD oligomerization/de-oligomerization is not a single-step process as it includes substrate binding, conformational rearrangements in the Ire1-LD dimer, and multi-step oligomerization events. The kinetics and thermodynamics of these steps depend on substrate and Ire1-LD concentrations and their interactions, but also on the parameters of the BiP chaperone cycle. For example, our unpublished data suggest that fine-tuning of BiP activity significantly affects the BiP efficiency to de-oligomerize Ire1-LD oligomers. However, while the understanding of this delicate and complicated balance is an extremely important and intriguing task that is likely

to require a single molecule approach. This is way beyond the scope of our current manuscript, but we will build on it, and it will be the focus of our future work.

Figure A. Turbidity measurements of 20 μM WT Ire1-LD after addition of 100 μM of ΔEspP (time 0); the Ire1-LD was incubated with the peptide for ca. 3 hours, then BiP (0, 0.2, 2 or 10 μM , as annotated) and 40 mM ATP were added.

References

Amin-Wetzel, N., Neidhardt, L., Yan, Y., Mayer, M.P., and Ron, D. (2019). Unstructured regions in IRE1alpha specify BiP-mediated destabilisation of the luminal domain dimer and repression of the UPR. *eLife* 8, e50793.

Belyy, V., Zuazo-Gaztelu, I., Alamban, A., Ashkenazi, A., and Walter, P. (2022). Endoplasmic reticulum stress activates human IRE1alpha through reversible assembly of inactive dimers into small oligomers. *eLife* 11.

Carrara, M., Prischi, F., Nowak, P.R., Kopp, M.C., and Ali, M.M. (2015). Noncanonical binding of BiP ATPase domain to Ire1 and Perk is dissociated by unfolded protein CH1 to initiate ER stress signaling. *eLife* 4.

Karagoz, G.E., Acosta-Alvear, D., Nguyen, H.T., Lee, C.P., Chu, F., and Walter, P. (2017). An unfolded protein-induced conformational switch activates mammalian IRE1. *eLife* 6.

Reviewer #2 (Comments to the Authors (Required)):

Summary:

ER stress plays critical roles in physiological and pathological events. The inositol-requiring enzyme 1 (IRE1) is one of the three widely recognized sensors of unfolded protein response (UPR). In this manuscript, Dawes et al., demonstrated that Bip regulated Ire1 by interacting in a ATP dependent manner. Furthermore, they identified 2 binding motifs of Ire1 interacting with Ire1. However, several significant concerns have been raised by reviewers:

Major Concerns:

1. This paper used a "clean" model consisting of proteins and peptides to study the interactions, but the interaction in cultured cells or in vivo will be more complicated. Some of the key findings should be evaluated in such a more physiological context.

The role of these Ire1 regions in Ire1 activation has been previously demonstrated *in vivo* (Ref 13 and 30). Particularly, it has been suggested that these regions are key for Ire1-LD oligomerization and its activity, highlighting the physiological importance of these regions for Ire1 function. However, the mechanism by which these regions control the Ire1 activation process remains elusive and is the focus of this work.

2. A concern raised from Fig.4b the most conserved scores including 306VVP308 are not within the 2 binding motif including 310GSTLPLL316.

The sequence conservation is likely to indicate the structural and/or functional importance of the corresponding region. While ³¹⁰GSTLPLL³¹⁶ is important for BiP binding (and thus highly conserved), ³⁰⁶VVP³⁰⁸ forms a short β -strand (Figure 4 in orange) and, thus, is likely to play an important structural role. Interestingly, this region, ³⁰⁶VVP³⁰⁸, is predicted by BiPPred to be a potential binding site but shows no interactions with BiP by NMR (Supplementary Figure 11), highlighting the importance of experimental validation of BiPPred predictions.

3. Fig2b left panel, To prove it's ATP depend. there should have control with no ATP but Δ EspP+different ratio of Bip. In the middle panel, the 2nd and 3rd lane indicating T229G and V461F mutations inactivate BiP, please explain how you inactivated the BiP for 1st.

We have showed that there is no de-oligomerization even in the presence of the highest BiP concentration (1:1) in the absence of ATP (Figure 2c). Not surprisingly, no de-oligomerization was observed at ten times lower BiP concentration (1:10); we now added a new Supplementary Figure 17 to demonstrate this. Several experiments in the manuscript

further validate that the Ire1-LD de-oligomerization process is ATP-dependent. Particularly, no de-oligomerization was observed if BiP cannot hydrolyse ATP (we used the ATP-deficient T229G variant to demonstrate this); no de-oligomerization was observed if BiP cannot bind to its canonical substrate (we used the V461F variant with a mutation near the substrate-binding site to demonstrate this). A brief description of the chaperone deficient BiP variants (including references to the original publications) is provided on page 6 of the manuscript; now we added more details about these variants in the figure legend as well.

4. There should be a loading control for WB in Fig. 2c and 2d. Showing the ratio of soluble and insoluble protein would be more straight to help support the conclusion.

Figure 2c and d show the SDS-PAGE gel rather than WB. The original images are available in the data depository. We now repeated the same experiment at a lower BiP concentration (new Supplementary Figure 17) and added lysozyme as a loading control to validate that the same amount of sample was loaded for each band.

5. Please add a list of abbreviations.

Thank you for the suggestion – these have been added.

6. Please add the statistic analysis statement in the methods

Thank you for the suggestion – these have been added.

Thoughtfully addressing these major and minor concerns will undoubtedly strengthen this review and enhance its potential impact in the field.

Reviewer #3 (Comments to the Authors (Required)):

In this study the researchers repeated the oligomerization of Ire1-LD induced by two reported model peptides, and based on this they found the size of Ire1-LD oligomers depends on peptide concentration and/or substrate affinity. The interesting finding is that insoluble Ire1-LD. Δ EspP oligomers have fibril-like morphology and folded-protein-like structure. Then the researchers found that Bip de-oligomerized the Ire1-LD. Δ EspP oligomers in an ATP-dependent way by interacting with two motifs in this flexible C-terminal region of Ire1-LD. Both motifs are near to the oligomerization interface of Ire1-LD and the region containing the two motifs are highly dynamics. So this study contributes some interesting and new findings to the community. However related deep digging is still not enough for these new findings. Some detailed considerations are as the following.

1. The structural feature of Ire1-LD was not described clearly, and in the structure of Ire1-LD. Δ EspP oligomers how the monomers arrayed is also not clear. Is there some evidence which showed that Δ EspP was in the oligomers? It is still a mystery how Δ EspP interacts with Ire1-LD, and then induces conformational change of Ire1-LD and its oligomerization. Although there is some conformational change of Ire1-LD upon binding of Δ EspP the researches still found Ire1-LD oligomer adopted its folded conformation. The binding ratio and mode between Ire1-LD and Δ EspP is unclear. The binding site in Ire1-LD of Δ EspP or the unfolded protein is also unknown. Do Δ EspP and the unfolded protein possibly share the same binding site in Ire1-LD?

We are also really intrigued by these challenging questions and are working in this direction; however, the key focus of this work is to investigate the role of BiP in Ire1-LD oligomerization and identify regions in Ire1 responsible for these interactions. Our results suggest that Δ EspP conformational rearrangements at the oligomerization interface promote both BiP binding and oligomerization, but the exact molecular details of these conformational changes are yet to be uncovered. See also our reply to Reviewer 1's comments.

What is the difference between the fibrillar structure of Ire1-LD. Δ EspP oligomers and amyloid fibrillar structure?

Ire1-LD oligomers have no characteristic structural features of amyloid fibrils (Supplementary Figure 8c); on the opposite to amyloids, the formation of Ire1-LD oligomers does not result in Ire1-LD unfolding (Supplementary Figure 8a) and Ire1-LD oligomers can be readily de-oligomerized by BiP (Figure 2). However, despite these differences, the Ire1-LD oligomerization process has some similarities with amyloid formation. Peptide binding results in conformational changes at the oligomerization interface of the Ire1-LD dimer and the formation of two 'sticky' ends (yellow in Figure 5) that can interact with other active Ire1-LD dimers (or oligomer). Consequently, similar to amyloid fibrils, the Ire1-LD oligomers extend by adding another Ire1-LD dimer (or oligomer) to their end. As a result, Ire1-LD oligomerization is a relatively slow process that depends on the concentrations of Ire1-LD 'active' (oligomerization-prone) dimers and oligomers.

2. It is reasonable that Bip cannot form stable interaction with Ire1-LD in the presence of ATP since ATP-state Hsp70 has lower affinity with substrate. In the presence of ADP or in the absence nucleotide, Hsp70 has higher affinity with substrate. However stable interaction between Bip and Ire1-LD is also not observed in the absence of ATP. In supplementary Figure 10 Bip was added into the labeled Ire1-LD oligomers to check their interaction. If the binding site in Ire1-LD is highly dynamics the NMR signal of this region will be very weak, and the signal change will not be so easy to be detected. However Ire1-LD dimer or oligomers were not added into labeled Bip to test the interaction as done for the two short peptides. It is interesting to find two Bip-binding peptides in the C-terminal region of Ire1-LD. However the affinity of their binding to Bip was not measured.

We would like to provide some clarifications for the experiments shown in Supplementary Figure 10, as we did exactly what the reviewer suggested above. Because only the flexible part of Ire1-LD is visible in NMR (Supplementary Figure 15), we used methyl-labelled BiP to monitor whether BiP could form an NMR-visible (relatively small) complex with unlabelled Ire1-LD in the presence and absence of ATP. Supplementary Figure 10 shows that no such complex was formed in the presence of ATP and within approximately 30 minutes after the addition of Δ EspP in the absence of ATP. Importantly, if BiP transiently interacts with larger Ire1-LD oligomers, it is not possible to detect these interactions by NMR for two reasons: (i) the size of the complex is too large (e.g., Figure 2a) to be directly observed by solution NMR; and (ii) transient interactions are unlikely to affect the peak positions/intensities of unbound-BiP (NMR visible). We also mentioned in the figure legend that, in the absence of ATP, BiP peaks gradually disappeared from the spectrum during a longer incubation (> 30 minutes), suggesting the formation of a larger (NMR invisible) complex over time, which is fully consistent with our other experiments. To demonstrate that BiP interacts with Ire1-LD oligomers, we used the SDS PAGE analysis (Figure 2c) that confirmed that in the absence of ATP BiP directly interacts with insoluble Ire1-LD aggregates.

The substrate promoted ATPase activity was only measured in one ratio of peptide: Bip (1:1?). It is better to observe elevated promotion with higher ratio of peptide: Bip.

We apologise for the lack of clarity. The ATPase activity measurements were performed for 1 μ M BiP in the presence of 1 mM of either peptide. The model HTFPAVL peptide binds to BiP with an affinity of \sim 10 μ M and, thus, under the experimental conditions, BiP is predominantly peptide-bound, resulting in only about 2-fold increase in ATPase activity; a very similar effect was observed for the two Ire1-derived BiP-binding peptides (Figure 3b). We now made this clear in the figure legend.

The effect of Bip binding to Ire1-LD. Δ EspP oligomers on the interaction between Ire1-LD and Δ EspP is unknown. Is there competition between binding of Bip and Δ EspP to Ire1-LD? Although it is found that the oligomerization interface and BiP binding sites of Ire1-LD is highly dynamic, it is unknown whether the dynamics changes with BiP binding, Δ EspP binding and Ire1-LD oligomerization.

We agree with the reviewer that the questions above are very important and intriguing, but we are not trying to address them in the current study. These will be the focus of our next study. The current work demonstrates that Δ EspP-induced conformational changes in Ire1-LD favour both oligomerization and BiP binding, enabling oligomerization of Ire1-LD (if BiP is not available) but also BiP-dependent control of this process (if BiP is available).

3. In the method about ATPase measurements, 1 mM of Hsp70 for ATPase assay is unusual. After malachite green reagent was added, 34% sodium citrate were added normally.

Thank for this. We corrected the typo in the Methods – 1 μ M of BiP was used for the MG assay. We used a standard commercial protocol for the assay and we now provided more details in the methods section to clarify.

4. In the method about protein purification, Hamster and human BiP were purified. However, it unclear when they are used in different experiments.

We have now added a data file to list the constructs and clarify which constructs were used for each experiment.

Although His tag will not affect ATPase activity it could affect substrate binding. So it should be careful if His tag was not removed in some experiments.

It has been demonstrated previously that the His-tag does not significantly affect BiP function (ref 28). Additionally, we showed that BiP can bind to Δ EspP in the presence and absence of the His tag as monitored by methyl NMR; moreover, BiP can de-oligomerize Ire1 in both the presence and absence of the His tag (Figure B).

Figure B. The His tag does not affect the BiP's ability to bind substrate and de-oligomerize Ire1-LD. (A) The percentage of domain docked conformation calculated based on the relative peak intensities for the three representative doublet peaks from (same conditions as for Supplementary Figure 11). Substrate binding results in the same conformational changes in the presence (BiP) and in the absence (BiP*) of the His-tag⁽ⁱ⁾. (B) The SDS-PAGE analysis of de-oligomerization of WT Ire1-LD in the presence and absence of BiP (left) and BiP* (right)⁽ⁱ⁾. 20 μ M Ire1-LD was incubated with 0 and 200 μ M Δ EspP for 3 hours. If required, 2 μ M BiP and 40 mM ATP were added to the reaction and incubated for another 3 hours, followed by collecting soluble and insoluble fractions.
⁽ⁱ⁾ BiP* contains a TEV-cleavable His tag, which was removed for the experiments shown in (A) and (B); BiP contains a non-cleavable His-tag (see Methods for more details).

5. Some small format problems, such as no blank between number and unit in some places (the last paragraph in P15) , missing subscript (H₂O in P15).

Thank you – this have been checked and corrected.

July 16, 2024

RE: Life Science Alliance Manuscript #LSA-2024-02702-TR

Dr. Anastasia Zhuravleva
University of Leeds
Leeds LS2 9JT
United Kingdom

Dear Dr. Zhuravleva,

Thank you for submitting your revised manuscript entitled "Chaperone BiP controls ER stress sensor Ire1 through interactions with its oligomers". We would be happy to publish your paper in Life Science Alliance pending final revisions necessary to meet our formatting guidelines.

- please be sure that the authorship listing and order is correct
- please upload all figure files individually, including the supplementary ones; all figure legends should only appear in the main manuscript file. Please remove the figures from the main manuscript file.
- please add your main, supplementary figure, and table legends to the main manuscript text after the references section. Please separate the Figure legends and Supplemental Figure legends into separate sections: "Figure legends" and "Supplementary figure legends."
- please label the Supplementary figures as Supplementary Figures S1, S2, etc.
- please incorporate the References listed in the Supplemental Material file into the main Reference list instead
- please upload your Tables in editable .doc or Excel format
- please add a Summary Blurb/Alternate Abstract to our system
- please add the Twitter handle of your host institute/organization as well as your own or/and one of the authors in our system
- title in the manuscript and system must match
- please upload a clean manuscript file without highlights in docx file format
- please use the [10 author names et al.] format in your references (i.e., limit the author names to the first 10)
- please add an Author Contributions section to your main manuscript text
- please add a callout for Figures S2A-B, S4A-D, S5A-B, S7A-C, S11A-C, S13A-B, S14A-B, S17A-B to your main manuscript text
- the link in the Data Availability statement does not appear to be working: <https://doi.org/10.5518/1533>

A. FINAL FILES:

-- Summary blurb (enter in submission system): A short text summarizing in a single sentence the study (max. 200 characters including spaces). This text is used in conjunction with the titles of papers, hence should be informative and complementary to the title. It should describe the context and significance of the findings for a general readership; it should be written in the

present tense and refer to the work in the third person. Author names should not be mentioned.

B. MANUSCRIPT ORGANIZATION AND FORMATTING:

Sincerely,

July 24, 2024

RE: Life Science Alliance Manuscript #LSA-2024-02702-TRR

Dr. Anastasia Zhuravleva
University of Leeds
Leeds LS2 9JT
United Kingdom

Dear Dr. Zhuravleva,

Thank you for submitting your Research Article entitled "Chaperone BiP controls ER stress sensor Ire1 through interactions with its oligomers". It is a pleasure to let you know that your manuscript is now accepted for publication in Life Science Alliance. Congratulations on this interesting work.

DISTRIBUTION OF MATERIALS:

Again, congratulations on a very nice paper. I hope you found the review process to be constructive and are pleased with how the manuscript was handled editorially. We look forward to future exciting submissions from your lab.

Sincerely,
